# Class I HDAC inhibitors enhance YB-1 acetylation and oxidative stress to block sarcoma metastasis

Amal M El-Naggar[1,2,3] (ID), Syam Prakash Somasekharan[4,†] (ID), Yemin Wang[2,†], Hongwei Cheng[4,†], Gian Luca Negri[5,†], Melvin Pan[2], Xue Qi Wang[2], Alberto Delaidelli[1,2], Bo Rafn[2], Jordan Cran[2], Fan Zhang[4], Haifeng Zhang[1,2], Shane Colborne[5], Martin Gleave[4] (ID), Anna Mandinova[6], Nancy Kedersha[7], Christopher S Hughes[2], Didier Surdez[8], Olivier Delattre[8], Yuzhuo Wang[4] (ID), David G Huntsman[1,2], Gregg B Morin[5] & Poul H Sorensen[1,2,*] (ID)

## Abstract

Outcomes for metastatic Ewing sarcoma and osteosarcoma are dismal and have not changed for decades. Oxidative stress attenuates melanoma metastasis, and melanoma cells must reduce oxidative stress to metastasize. We explored this in sarcomas by screening for oxidative stress sensitizers, which identified the class I HDAC inhibitor MS-275 as enhancing vulnerability to reactive oxygen species (ROS) in sarcoma cells. Mechanistically, MS-275 inhibits YB-1 deacetylation, decreasing its binding to 5′-UTRs of *NFE2L2* encoding the antioxidant factor NRF2, thereby reducing *NFE2L2* translation and synthesis of NRF2 to increase cellular ROS. By global acetylomics, MS-275 promotes rapid acetylation of the YB-1 RNA-binding protein at lysine-81, blocking binding and translational activation of *NFE2L2*, as well as known YB-1 mRNA targets, *HIF1A*, and the stress granule nucleator, *G3BP1*. MS-275 dramatically reduces sarcoma metastasis *in vivo*, but an MS-275-resistant YB-1K81-to-alanine mutant restores metastatic capacity and NRF2, HIF1α, and G3BP1 synthesis in MS-275-treated mice. These studies describe a novel function for MS-275 through enhanced YB-1 acetylation, thus inhibiting YB-1 translational control of key cytoprotective factors and its pro-metastatic activity.

**Keywords** HDAC inhibitors; metastasis; NRF2; sarcoma; YB-1
**Subject Categories** Cancer; Metabolism; Post-translational Modifications & Proteolysis

## Introduction

Tumor metastasis is widely held to be an inefficient process, with only a tiny fraction of primary tumor cells surviving the many steps of the metastatic process to form distant metastases [1,2]. Metastatic dissemination is highly stressful, with each step continuously subjecting tumor cells to heterogeneous microenvironments and diverse stresses to potentially cull pre-metastatic cells [3]. For example, oxidative stress and accumulation of reactive oxygen species (ROS) prevents melanoma metastasis in cells unable to adapt by mounting an efficient antioxidant response [4,5]. ROS levels are tightly controlled in tumor cells to prevent ROS-induced toxicity [6], in part through activation of the master antioxidant transcription factor, NRF2 [7]. Activation of NRF2 is largely attributed to its protein stabilization under oxidative stress, due to dissociation from its repressor protein Keap1 when the latter becomes oxidized [8], or through increased transcription of nuclear factor erythroid 2-related factor 2 (*NFE2L2*) encoding NRF2 [9]. For high-risk childhood bone sarcomas including Ewing sarcoma (EwS) and osteosarcoma (OS), a major roadblock to improving outcomes is to prevent or treat metastatic disease. Indeed, metastasis is the single most powerful predictor of outcome in these diseases, but essentially no progress has been made for the past several decades in improving survival of sarcoma patients with metastatic disease. We therefore investigated how adaptation to oxidative stress might facilitate metastasis in EwS and OS, and if this process could be targeted therapeutically.

Tumors are continually exposed to multiple forms of stress, including oxidative stress, hypoxia, nutrient depletion, genotoxic stress, and cytotoxic therapy. Each is potentially lethal unless tumor cells can acutely adapt to it. Stress adaptation via mutationally

---

1 Department of Pathology & Laboratory Medicine, University of British Columbia, Vancouver, BC, Canada
2 Department of Molecular Oncology, BC Cancer, part of the Provincial Health Services Authority, Vancouver, BC, Canada
3 Department of Pathology, Faculty of Medicine, Menoufia University, Shibin El Kom, Egypt
4 Vancouver Prostate Centre, Vancouver, BC, Canada
5 Michael Smith Genome Sciences Centre, Vancouver, BC, Canada
6 Brigham and Women's Hospital, Harvard University, Boston, MA, USA
7 Massachusetts General Hospital, Harvard University, Boston, MA, USA
8 Centre de recherche de l'Institut Curie, Paris, France
*Corresponding author. Tel: +1 604 675 8202; E-mail: psor@mail.ubc.ca
†These authors contributed equally to this work

driven clonal selection is postulated to underlie acquisition of aggressive phenotypes including chemoresistance and metastatic capacity [10]. However, accumulating evidence, including our own work [11], suggests that stress adaptation also occurs through acute changes in mRNA translation and protein synthesis [12]. For example, under hypoxia, translation of pro-growth mRNAs is largely inhibited, while that of mRNAs encoding HIF1α and other stress proteins is enhanced to promote survival of hypoxic tumor cells [13]. Similarly, ER stress initiates the unfolded protein response, which inhibits global translation through phosphorylation of the ternary complex component, eIF2α, by at least four stress activated kinases, but with selective translation of proteins such as BIP and chaperones key for cell survival [14]. Selective translation of key cytoprotective factors in such settings allows tumor cells to rapidly respond to changing microenvironments without the need for protracted transcriptional responses [15]. Recent work suggests that translational reprogramming is particularly important for survival of tumor cells exposed to increased oxidative stress. For example, haploinsufficiency for the major mRNA cap binding protein, eIF4E, significantly impedes cellular transformation and deficiency in translation of mRNA that mitigate oxidative stress [16]. Moreover, pancreatic carcinoma cells with loss of NRF2 show defects in redox homeostasis and markedly diminished tumor initiation and maintenance, which is linked to translational inhibition due to oxidation of the different members of the translation machinery [17]. Therefore, a greater understanding of how translation regulates redox homeostasis may uncover new strategies for targeting metastatic disease.

One factor known to function in translational control of stress-adaptive responses is Y-box binding protein 1 (YB-1/YBX1). YB-1 is an RNA-binding protein (RBP) that binds to 5′- and 3′-untranslated regions (UTRs) of mRNAs mainly through its highly conserved cold shock domain (CSD) [18]. This protein is highly expressed in both EwS and OS, where it is strongly associated with poor outcome [19,20]. YB-1 translationally activates diverse stress response factors with pro-metastatic activities in human malignancies. In breast cancers, YB-1 translationally controls the epithelial-to-mesenchymal transition (EMT) by activating expression of transcription factors such as SNAIL, TWIST, and ZEB2 to drive breast cancer EMT and metastasis [21]. In colorectal carcinoma metastasis, YB-1 promotes liver metastasis by translationally regulating the IGF1 receptor [22]. In sarcomas, YB-1 facilitates metastasis by directly binding the *HIF1A* 5′-UTR to activate its translation and increase HIF1α synthesis under hypoxia [19]. Other potential pro-metastatic functions include roles in stabilizing oncogenic transcripts [23], binding of tRNA fragments to mediate cytoprotective oxidative stress-induced translational repression [24], and translational activation of the Rho GTPAse-dependent ROCK1 ser/thr kinase to increase cell motility [25]. We also found that in sarcomas, YB-1 binds and activates mRNA encoding Ras-GTPase-activating protein (SH3 domain) binding protein 1 (G3BP1), a key stress granule (SG) nucleating protein [26,27]. SGs, mainly studied under oxidative stress, are cytoplasmic aggregates composed of RBPs, the 40S ribosome, stalled translation initiation complexes, and silenced mRNAs that form rapidly under cell stress, and recent studies have begun to uncover the composition of these structures [28–31]. YB-1 is essential for *G3BP1* translation and SG formation in sarcomas, and G3BP1 deficiency leading to loss of SGs blocks metastatic capacity in EwS and OS [32]. We hypothesize that by

mediating these diverse stress responses, YB-1 confers increased fitness to tumor cells.

In an effort to uncover new strategies to target metastatic disease in EwS and OS, we performed small molecule screens to search for agents that induce ROS accumulation and oxidative stress in sarcoma cell lines, with the rationale being that such compounds should further decrease the efficiency of, and therefore limit, metastasis in these diseases. Our studies unexpectedly highlight class I histone deacetylase (HDAC) inhibitors, including MS-275 (Entinostat), as increasing ROS in sarcoma cells and blocking their capacity for invasion and metastasis *in vivo*. Mechanistically, MS-275 induces acetylation within the YB-1 CSD to block binding of target mRNAs, with widespread effects on YB-1 translational regulation of stress-adaptive mRNAs, including a new YB-1 target, *NFE2L2*, encoding NRF2. These studies therefore have important implications for therapeutic targeting of metastasis in childhood bone sarcomas.

## Results

### Cell-based screens identify MS-275 as an oxidative stress sensitizer in sarcoma cells

Oxidative stress blocks melanoma metastasis [4,5], leading to the notion that agents enhancing ROS accumulation might be effective as anti-metastatic therapeutics. However, tumor cells can adapt and reduce ROS by activating a variety of antioxidant stress response pathways [33]. We therefore reasoned that an alternative approach might be to block such adaptive responses, rather than by increasing ROS itself. To evaluate this, we established a cell-based screen for compounds that increase sensitivity to oxidative stress by inducing apoptosis in high-risk human sarcoma cells, but only when combined with sublethal doses of sodium arsenite (NaAsO$_2$), commonly used to stimulate ROS production in cell culture [34]. We initially focused on epigenetic regulators, given that both EwS [35] and OS [36] are strongly linked to epigenetic deregulation, and because many compounds in this class are already approved for human use. We used a commercial library of 92 compounds directed against HDACs, histone acetyltransferases, methyltransferases, demethylases, and acetyl-lysine readers. U2OS OS cells were pre-treated for 24 h with the above library at 1 μM, followed by a 1-h treatment with a sublethal dose of 100 μM NaAsO$_2$, followed by additional 12-h treatment with the above library. Nine compounds significantly enhanced NaAsO$_2$-induced death (Fig 1A and Appendix Table S1); three were class I HDAC inhibitors, MS-275 (Entinostat), Quisinostat, and Romidepsin, and two were pan-HDAC inhibitors, Panobinostat, and SAHA (Appendix Table S1). Of these compounds, only MS-275 had no toxicity alone at a 1 μM concentration (Fig 1B) and did not affect cell proliferation (Fig EV1A); however, MS-275-pre-treated cells were potently killed after a subsequent 1-h exposure to 100 μM NaAsO$_2$, with a greater effect than NaAsO$_2$ alone (Fig 1B), so we focused on MS-275 in subsequent studies. MS-275 is a synthetic benzamide derivative HDACi, which preferentially targets HDAC1 and HDAC3 [37].

We first confirmed that MS-275 increases oxidative stress in NaAsO$_2$-treated sarcoma cells. Using chloromethyl derivative of 2′,7′-dichlorofluorescin diacetate (CM-H$_2$DCFDA) staining as a ROS readout, MS-275 alone failed to significantly increase ROS levels

    

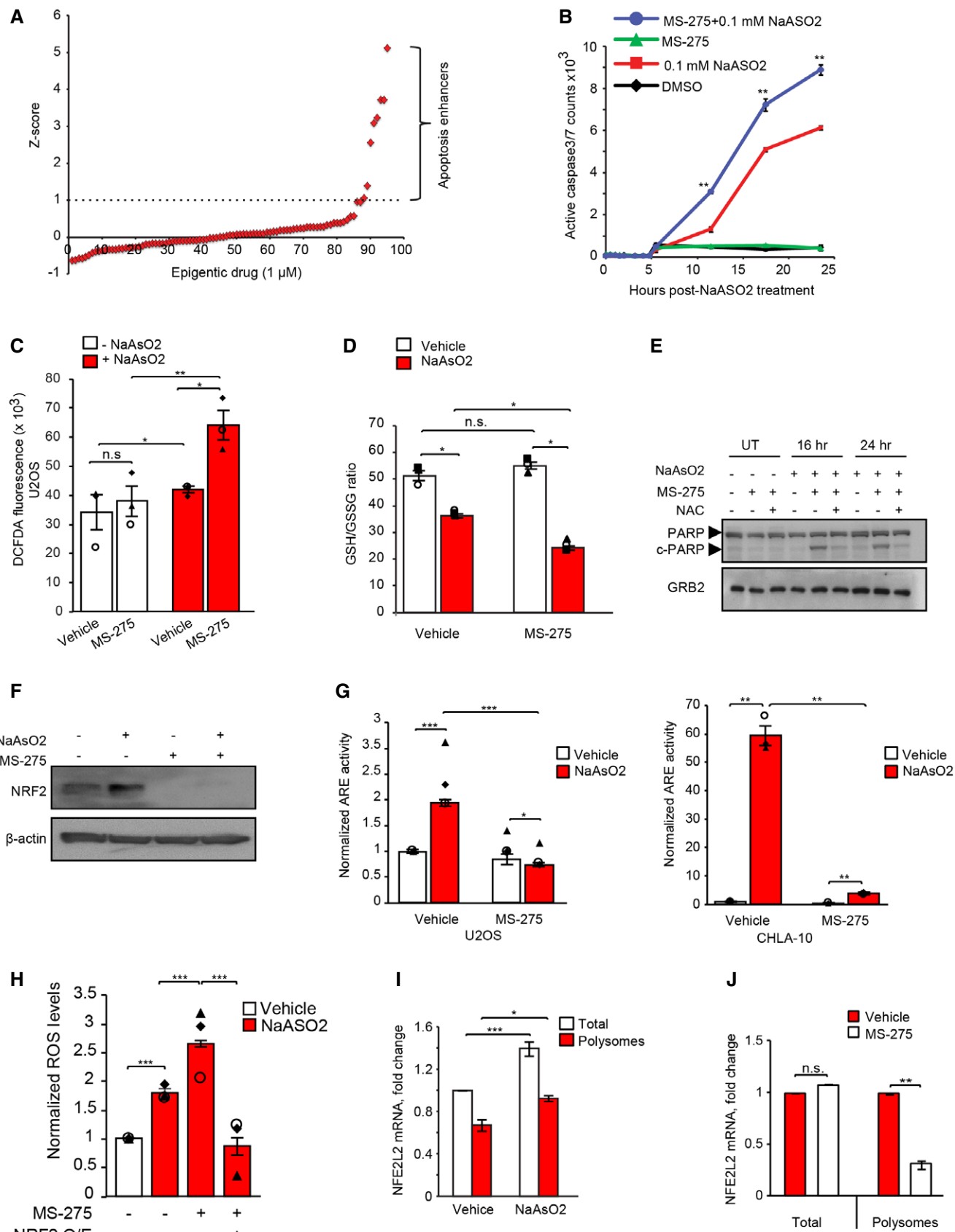

Figure 1.

**Figure 1. Class 1 HDAC inhibitors are potent oxidative stress sensitizers in sarcoma cells.**

A  Cell-based screens for ROS sensitizers in U2OS cells (see Materials and Methods). Average $z$-scores for activated caspase-3/7 from three independent experiments are plotted.

B  Time course of treatment with 1 μM class I HDAC inhibitor MS-275 for 24 h alone or in combination with 100 μM $NaAsO_2$ for 1 h to assess U2OS viability by NucView-488 fluorescence. DMSO was used as a vehicle control. Error bars indicate SEM ($n$ = 3 independent experiments, each performed in triplicate).

C  U2OS cells were treated without (vehicle) or with MS-275 (1 μM, 24 h), and without ($-NaAsO_2$) or with $NaAsO_2$ ($+NaAsO_2$) (500 μM, 1 h) and assessed for ROS levels using $CM-H_2DCFDA$. ROS levels were normalized to protein content. Error bars indicate SEM for $n$ = 3 independent experiments, each performed in triplicate.

D  U2OS cells +/− MS-275 (1 μM, 24 h) were exposed to vehicle (white bars) or $NaAsO_2$ (red bars; 500 μM, 1 h) and assessed for redox stress using intracellular GSH/GSSG ratios as a readout. Data are expressed as mean ± SEM. for $n$ = 3 independent experiments, each performed in triplicate.

E  Immunoblotting showing effects of the antioxidant N-acetylcysteine (NAC) on PARP cleavage at the indicated time points in U2OS cells subjected to treatment with or without MS-275 (1 μM, 24 h) and $NaAsO_2$ (100 μM, 1 h). GRB2 was used as a loading control.

F  Immunoblotting showing effects of MS-275 on NRF2 expression in U2OS cells treated with $NaAsO_2$ (500 μM, 1 h) as indicated. β-actin was used as a loading control.

G  Luciferase reporter assays showing antioxidant response element (ARE) activity in U2OS cells (left panel) and CHLA-10 cells (right panel) in the presence or absence (vehicle) of MS-275 (1 μM, 24 h) and either untreated (UT; white bars) or treated with $NaAsO_2$ (100 μM, 1 h; red bars). Error bars indicate SEM for $n$ = 3 independent experiments, each performed in triplicate.

H  U2OS was transfected with empty vector or NRF2-expressing vector. Two days post-transfection, cells were treated without (vehicle) or with MS-275 (1 μM, 24 h), and without ($-NaAsO_2$) or with $NaAsO_2$ ($+NaAsO_2$) (100 μM, 1 h) and assessed for ROS levels using $CM-H_2DCFDA$. Data are presented as fold change over vehicle control ($-NaAsO_2$) for each group. Error bars indicate SEM for $n$ = 3 independent experiments, each performed in triplicate.

I  Total RNA (white bars) and polysome fractionated RNA (red bars) were isolated from U2OS cells treated +/− $NaAsO_2$ (100 μM, 1 h), and assayed for *NFE2L2* expression by RT–PCR using *NFE2L2* primers. Data were normalized against *GAPDH* and expressed as fold change ± SEM of two independent experiments, each performed in triplicate.

J  Total and polysomal RNA from U2OS cells treated without (vehicle) or with MS-275 (1 μM, 24 h) and further treated with $NaAsO_2$ (100 μM, 1 h) were subjected to polysomal fractionation followed by RT–PCR using primers for *NFE2L2*. Data were normalized against *GAPDH* and expressed as fold change ± SEM of 2 independent experiments, each performed in triplicate.

Data information: Unpaired two-tailed Student's $t$-test, $*P < 0.05$; $**P < 0.005$; $***P < 0.0005$; n.s. = non-significant.
Source data are available online for this figure.

compared to vehicle in U2OS cells (Figs 1C and EV1B), and only moderately increased ROS in CHLA-10 cells (Fig EV1C). However, MS-275 markedly increased ROS in combination with $NaAsO_2$ co-treatment, in U2OS (Figs 1C and EV1B) and CHLA-10 EwS cells (Fig EV1C), which was confirmed using CellROX assays (Fig EV1D). Moreover, MS-275 significantly decreased the ratio of reduced to oxidized forms of glutathione (GSH/GSSG), a well-established readout of oxidative stress in cells and tissues, but only in combination with $NaAsO_2$ (Fig 1D), and only increased apoptosis in $NaAsO_2$-co-treated U2OS cells, which was reversed by the antioxidant, N-acetylcysteine (NAC; Fig 1E). We next cultured sarcoma cells as non-adherent suspension cultures, since cancer cell detachment from the extracellular matrix, a putative initial step of metastasis [38], increases intracellular ROS [39]. Transfer to suspension cultures led to rapid formation of sarcoma spheroids, as described [40]; this dramatically increased ROS levels, which was reversed by NAC or catalase antioxidants (Fig EV1E). Treatment of sarcoma spheroids with MS-275 further increased ROS, which again could be reversed by antioxidants (Fig EV1F), and increased apoptosis, which was also blocked by antioxidants (Fig EV1G). Therefore, MS-275 increases ROS accumulation and oxidative stress in $NaAsO_2$-exposed bone sarcoma cells.

**MS-275 non-transcriptionally blocks NRF2 synthesis under oxidative stress**

We next investigated how MS-275 regulates ROS homeostasis in sarcoma cells. ROS levels are tightly controlled in cells to prevent ROS-induced toxicity [6], in large part through induction of the NRF2 master antioxidant transcription factor [7]. We therefore expected that MS-275 treatment would increase NRF2 expression in response to enhanced ROS accumulation. Surprisingly, however, while $NaAsO_2$ robustly induced NRF2 protein levels in U2OS cells

(Fig 1F), this was completely blocked by MS-275 treatment, either with or without $NaAsO_2$ co-treatment (Fig 1F). Moreover, while $NaAsO_2$ enhanced NRF2-linked antioxidant response element (ARE) reporter activity, this was again significantly reduced by MS-275 in U2OS and CHLA-10 cells (Fig 1G, left and right panels, respectively). Lastly, ectopic NRF2 expression blocked the ability of MS-275 to induce ROS accumulation in sarcoma cells under $NaAsO_2$ co-treatment (Fig 1H).

We next wished to determine how MS-275 influences NRF2 expression. HDAC inhibitors such as MS-275 are thought to transcriptionally modify gene expression through hyperacetylation of histones such as H3 and H4, leading to chromatin remodeling which typically requires ~24 h or more to cause major transcriptional changes [41]. We therefore first measured levels of *NFE2L2* transcripts (encoding NRF2) and found that total levels and those enriched in polysomes isolated from sucrose gradients (i.e., translating mRNAs bound to ribosomes; [42]) were increased by $NaAsO_2$ (Fig 1I), suggesting that $NaAsO_2$ increases both transcription and translation of *NFE2L2*. Unexpectedly, however, total *NFE2L2* levels were unchanged when cells were treated with MS-275; instead, this agent markedly reduced the fraction of *NFE2L2* associated with polysomes (Fig 1J). This suggests that MS-275 specifically inhibits *NFE2L2* translation rather than its transcription (see below). We further analyzed the polysome profiles of U2OS cells, demonstrating that while MS-275 fails to reduce polysome formation (i.e., does not block translation), $NaAsO_2$ does so (i.e., reduced polysomes and increased 80S formation), regardless of MS-275 addition (Fig EV1H). We also tested whether MS-275 influences NRF2 protein stability by performing cycloheximide pulse-chase experiments in MS-275-treated U2OS and CHLA-10 cells. However, MS-275 failed to reduce NRF2 stability in either cell line (Fig EV1I and J). Together, these results suggest that the major effect of MS-275 on NRF2 activity is by reducing its mRNA translation.

## MS-275 enhances acetylation of the YB-1 translational activator in sarcoma cells

The above findings indicate that MS-275 regulates NRF2 via mechanisms other than through histone modifications. In addition to histones, HDAC inhibitors also block deacetylation of other proteins such as p53 and Rb [43]. To perform an unbiased screen for other potential MS-275 acetylation targets in sarcoma cells, lysates from MS-275/NaAsO$_2$-treated CHLA-10 cells were subjected to immunoprecipitation (IP) using anti-acetyl-lysine (α-acK) antibodies [44] and then processed for tandem mass spectrometry (MS/MS). This identified 1,544 α-acK affinity-purified proteins enriched in response to MS-275 (Dataset EV1). From this list, we specifically looked for proteins involved in mRNA translation, such as members of the translation machinery and RNA-binding proteins, given the observed effects of MS-275 on *NFE2L2* translation. This identified a number of known translation-associated proteins (Figs 2A and EV2A), but of these, only the YB-1 RBP was consistently enriched by α-acK after MS-275 treatment as were a number of histones and several other proteins. Indeed, apart from histones, and several ribosomal proteins, only DEK (DEK proto-oncogene), NAT10 (N-acetyltransferase 10), PARP1 (poly(ADP-ribose) polymerase 1), and TOP1 (DNA topoisomerase I), in addition to YBX1, showed consistently increased acetylation under MS275 treatment. YB-1 was of particular interest because of our previous work showing that YB-1 translationally activates other stress-adaptive mRNA in tumor cells under stress [19,21,32]. YB-1 acetylation was validated by subjecting lysates from MS-275/NaAsO$_2$-treated U2OS cells to IPs using the same α-acK antibodies [44], followed by YB-1 immunoblotting (Fig EV2B). Treatment with MS-275 had no effects on total YB-1 protein levels. There was no detectable acetylation of other RBPs such as G3BP1 (Fig EV2B). We then performed a time course analysis of YB-1 acetylation which, remarkably, showed that YB-1 acetylation increased after as little as 30 min of MS-275 treatment (Fig 2B). Histone H4, used as a positive control, was also acetylated in the presence of MS-275 but over a much longer time course; H4 acetylation was minimal at 3 h in these cells, but then increased markedly by 6–24 h of MS-275 treatment (Fig EV2C). Furthermore, other class I HDAC inhibitors including Quisinostat and Romidepsin at 1 uM also resulted in enhanced YB-1 acetylation, similar to MS-275 (Fig EV2D).

These data indicate that MS-275 enhances acetylation of YB-1 in sarcoma cells. To further explore this, YB-1 was IP'ed from U2OS cells using anti-YB-1 antibodies, followed by MS/MS analysis. This identified a major acetylated YB-1 lysine, namely K81, which has previously been reported in global acetylome studies [44]. Several other minor lysine-acetylated peptides were identified in our studies, but these could not be reproducibly detected. We therefore generated a K-to-alanine (A) substitution of K81 and expressed this mutant along with wild-type (wt) YB-1 as FLAG-tagged proteins in U2OS cells (Fig EV2E). Cell lysates were assessed for YB-1 acetylation after 3 h of MS-275 treatment (to minimize potential secondary effects of drug treatment). This demonstrated robust *wt* YB-1 acetylation in the presence of MS-275, as well as of histone H4 control (Fig 2C). Comparatively, however, YB-1-K81A acetylation was markedly reduced, either with or without MS-275 treatment, suggesting that K81 is a major site for YB-1 acetylation in the presence of MS-275.

## YB-1 translationally activates *NFE2L2*, which is blocked by MS-275

We next wished to determine whether YB-1 acetylation is linked to effects of MS-275 on NRF2 expression. We previously showed that YB-1 inactivation increases sensitivity to oxidative stress in sarcoma cells [32]. We therefore wondered whether YB-1 levels directly affect NRF2 expression. Indeed, YB-1 knockdown under H$_2$O$_2$- or NaAsO$_2$-induced oxidative stress markedly reduced NRF2 levels *in vitro* (Fig 3A). Given that YB-1 functions as a translational activator of diverse stress-adaptive mRNAs such as HIF1α and G3BP1 [19,32], we next tested whether YB-1 also controls *NFE2LE* translation. First, we used a bicistronic reporter containing *NFE2L2* 5-UTR sequences required for *NFE2L2* translation [45] linked to Firefly luciferase (LUC), as well as Renila LUC driven by the 5′-UTR of *HBB* (beta-globin). Using this reporter in cell-free transcription/translation studies, recombinant YB-1 markedly enhanced Firefly LUC translation in a dose-dependent manner, but not of Renila LUC; therefore, YB-1 translationally activates the *NFE2L2* 5′-UTR but not a control *HBB* 5′-UTR (Fig 3B). Next, using electrophoretic mobility shift assays (EMSAs) for YB-1 binding as described [32], we found that YB-1 directly bound, in a concentration-dependent manner, to labeled *NFE2L2* 5′-UTR probe (i.e., by gel retardation), which was completely blocked by unlabeled probe (Fig 3C). This shows unambiguously that YB-1 binds the *NFE2L2* 5′-UTR.

Next, since K81 maps within the RNA-binding CSD of YB-1 [46], we wondered whether MS-275-induced YB-1 K81 acetylation modifies YB-1 translational control of NRF2. To test this, we assessed whether K81 acetylation specifically inhibits binding of YB-1 to *NFE2L2* to reduce its translation. We therefore ectopically expressed FLAG-tagged *wt* YB-1 versus YB-1-K81A in CHLA-10 cells and performed RNA immunoprecipitation (RIP) with anti-FLAG antibodies followed by qRT–PCR to determine relative *NFE2L2* binding under oxidative stress to each protein +/− MS-275 treatment. YB-1-K81A bound significantly higher amounts of *NFE2L2* than *wt* YB-1 under all conditions tested, and especially with NaAsO$_2$ and MS-275 co-treatment (Fig 3D). To confirm that MS-275 is specifically affecting YB-1-mediated translation of *NFE2L2*, we monitored acute synthesis of NRF2 under NaAsO$_2$ and MS-275 treatment, using L-azidohomoalanine (AHA) Click chemistry as described [47]. Briefly, cells were pulsed briefly with AHA, such that only newly synthesized AHA-labeled proteins become biotinylated and affinity captured with streptavidin beads, and newly synthesized proteins can then be analyzed by immunoblotting. As shown in Fig 3E, NaAsO$_2$ induced robust new synthesis of NRF2 in *wt* YB-1-expressing cells (compare lanes 5–6), but this was completely blocked in MS-275-co-treated cells (compare lanes 6–7). However, NRF2 synthesis was rescued in cells expressing YB-1-K81A (Fig 3E; compare lanes 4–7). This strongly indicates that cells expressing a form of YB-1 insensitive to MS-275 acetylation retain acute *NFE2L2* translation capacity even in the presence of drug. Together, our findings strongly support a model whereby MS-275 increases acetylation of YB-1-K81 within the YB-1 CSD to inhibit *NFE2L2* binding, in turn blocking its translational activation by YB-1.

## MS-275 blocks YB-1 translational control of *G3BP1* and *HIFA* mRNAs via K81 acetylation

We next tested whether effects of MS-275 on *NFE2L2* translation extend to other known YB-1 mRNA targets, namely *G3BP1* [32] and

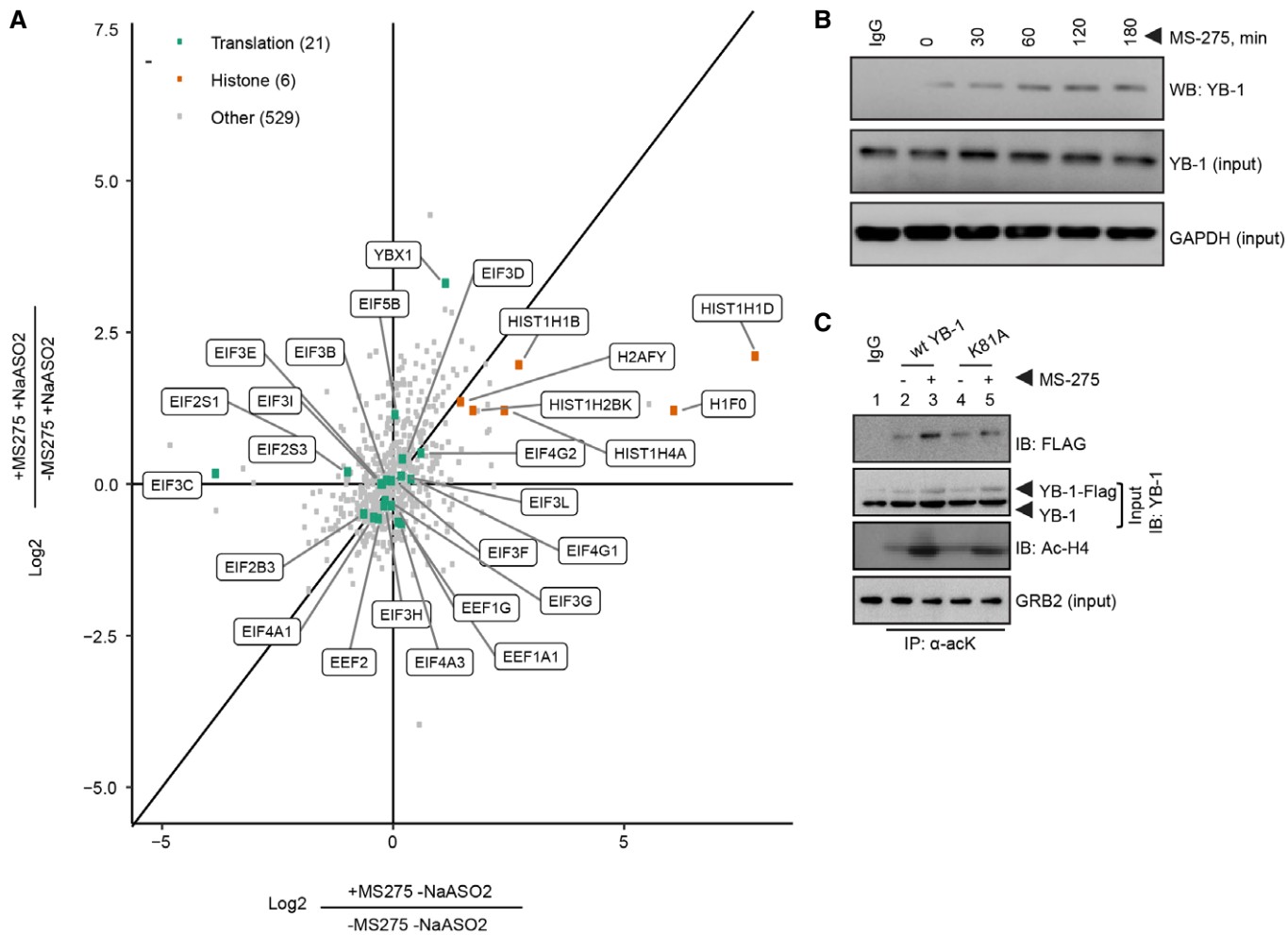

**Figure 2. MS-275-induced YB-1 acetylation.**

A Scatterplot showing log2 transformed ratios of heavy amino acid (MS-275 treated; +MS-275) over light amino acid (vehicle; −MS-275) conditions with the indicated NaAsO$_2$ treatment (250 μM, 1 h). Dots are color coded to identify translation-associated proteins (green) histones (orange) or other categories of proteins (gray).

B Analysis of YB-1 acetylation in U2OS cells +/− MS-275 (1 μM) over the indicated time course. Lysine-acetylated proteins were affinity purified using anti-acetyl-lysine (α-acK) antibodies and analyzed by immunoblotting using antibodies to YB-1. IgG was used as negative antibody control and GAPDH as a loading control.

C Differential acetylation of FLAG-tagged *wt* YB-1 or the indicated FLAG-tagged lysine (K) to alanine (A) YB-1 mutant (K81A) in the presence of MS-275. U2OS cells were transfected with FLAG-tagged *wt* YB-1 or FLAG-tagged K81A. Cells were then treated with vehicle alone or MS-275 (1 μM, 24 h). Lys-acetylated proteins were immunoprecipitated using α-acK antibodies, and acetylated FLAG-tagged proteins were analyzed by anti-FLAG, YB-1, or acetylated histone H4 (Ac-H4). IgG was used as a negative antibody control.

Source data are available online for this figure.

HIF1A [19]. In CHLA-10 cells, MS-275 markedly reduced both HIF1α under hypoxia (Fig 4A), and G3BP levels under NaAsO$_2$ treatment (Fig 4B), without affecting YB-1 levels. Similar to *NFE2L2*, association of either *HIF1A* (encoding HIF1α) or *G3BP1* transcripts (encoding G3BP) with polysomes was significantly reduced in MS-275-treated cells (Fig 4C and D), even though total mRNAs were increased. Using RIP with anti-FLAG antibodies as described above for *NFE2L2*, YB-1-K81A bound significantly higher amounts of *HIF1A* and *G3BP1* transcripts compared to *wt* YB-1, especially under MS-275 co-treatment with 1% O$_2$ or NaAsO$_2$, respectively (Fig 4E and F). Again, MS-275 completely blocked acute G3BP1 synthesis under NaAsO$_2$, or HIF1α under 1% O$_2$, both of which were rescued by YB-1-K81A but not *wt* YB-1 (Fig EV3A and B).

Finally, as a functional readout of MS-275 activity, we compared U2OS cells overexpressing *wt* YB-1 or YB-1K81A and treated these cells with arsenite +/− MS-275 to determine effects on SG formation. We used cells overexpressing *wt* G3BP1 as a positive control, as G3BP1 overexpression induces SGs even in the absence of stress [48]. While *wt* YB-1 was unable to rescue SGs in the presence of MS-275, YB-1-K81A was almost as effective as G3BP1 overexpression in restoring NaAsO$_2$-induced SG assembly in MS-275-treated cells (Fig EV3D). Therefore, MS-275-enhanced acetylation of YB-1 K81 is directly associated with its ability to block YB-1-mediated translational activation of target mRNAs and SG formation.

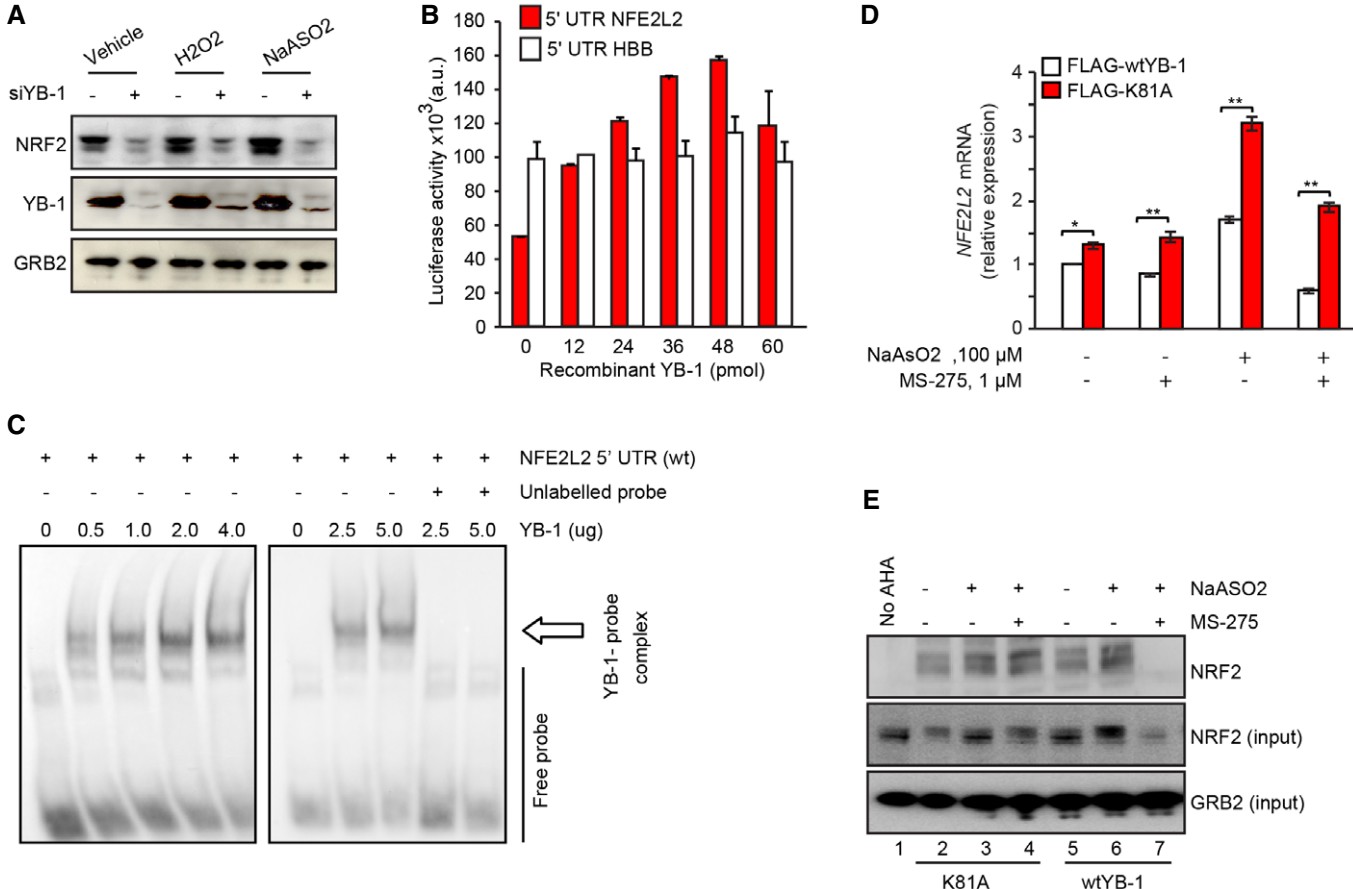

**Figure 3. NRF2 is a novel YB-1 downstream target.**

A   Immunoblotting showing effects of YB-1 KD on NRF2 expression in U2OS cells under the indicated oxidative stress inducing conditions: NaAsO$_2$ (100 μM, 1 h) and H$_2$O$_2$ (200 μM, 1 h). GRB2 was used as a loading control

B   *In vitro* cell-free translation assay using reporter constructs with *NFE2L2* or *beta-globin* (*HBB*) 5′-UTRs linked to the SP6 RNA polymerase promoter incubated with increasing amounts of recombinant YB-1 protein and assessed for LUC activity. Results are displayed as means ± SD from two independent experiments, each performed in triplicate.

C   Electrophoretic mobility gel shift assay (EMSA) to detect binding of recombinant YB-1 to the *NFE2L2* 5′-UTR. Biotin end-tagged *NFE2L2* 5′-UTR probe (Bio-UTP-*NFE2L2* 5′-UTR) and unlabeled full-length *NFE2L2* 5′-UTR were incubated with recombinant GST-YB-1. Arrow shows supershifted Bio-UTP-RNA probe/YB-1 complexes

D   *NFE2L2* mRNA binding to FLAG-tagged wt or YB-1-K81A in CHLA-10 cells +/− MS-275 (1 μM, 24 h) and NaAsO$_2$ (100 μM, 1 h), as measured by qRT–PCR of total RNA isolated from cell lysates. Data were normalized for each sample against the geometric mean of *YBX1* mRNA binding and expressed as fold change ± SEM of two independent experiments, each performed in triplicate. Unpaired two-tailed Student's *t*-test;*P < 0.05; **P < 0.005.

E   Acutely synthesized NRF2 in CHLA-10 cells expressing wtYB-1-FLAG or YB-1-K81A-FLAG. Cells were treated +/− MS-275 (1 μM, 2 h). Then, cells were methionine starved and then pulsed with AHA and NaAsO$_2$ (100 μM), with continuation of MS-275 treatment for 1 h. Acutely synthesized NRF2 was identified by immunoblotting with NRF2 antibodies (top blot) and compared to total NRF2 levels (middle blot). Total GRB2 was used as a loading control (lower blot).

Source data are available online for this figure.

## MS-275 blocks bone sarcoma metastasis *in vivo*

Given the ability of MS-275 to induce oxidative stress in sarcoma cells, we directly tested whether MS-275 influences sarcoma metastatic capacity *in vivo*. We used the well-established murine renal subcapsular implantation model [49], which we previously exploited to study sarcoma metastasis [19,50]. CHLA-10 EwS cells were implanted under the renal capsules of immunocompromised mice, and mice were treated 3 weeks post-inoculation with orally administered vehicle or MS-275 at 20 mg/kg, 5 days/week for 3 weeks and monitored for tumor growth. At endpoint, there were no significant differences in proliferative rates between vehicle- and MS-275-treated tumors. We therefore analyzed MS-275 effects on

invasive and metastatic capacity of CHLA-10 tumor xenografts. Vehicle-treated CHLA-10 tumors (Fig EV4A panel I and B) showed highly infiltrative borders with direct invasion into adjacent normal kidney, as previously reported for these cells [19,50]. In stark contrast, MS-275-treated CHLA-10 tumors showed non-invasive "pushing" borders that failed to penetrate into neighboring normal kidney (Fig EV4A panel ii and B). Therefore, MS-275 inhibits EwS invasion *in vivo*. Moreover, using H&E morphology (Fig EV4A, panels *iii-iv*) and immunohistochemistry (IHC) for the EwS marker CD99 (Fig EV4A, panels *v-vi*), we observed a dramatic block in metastatic spread to lungs in MS-275-treated mice (Fig EV4C). Therefore, MS-275 inhibits local invasion and metastatic capacity of EwS cells *in vivo*.

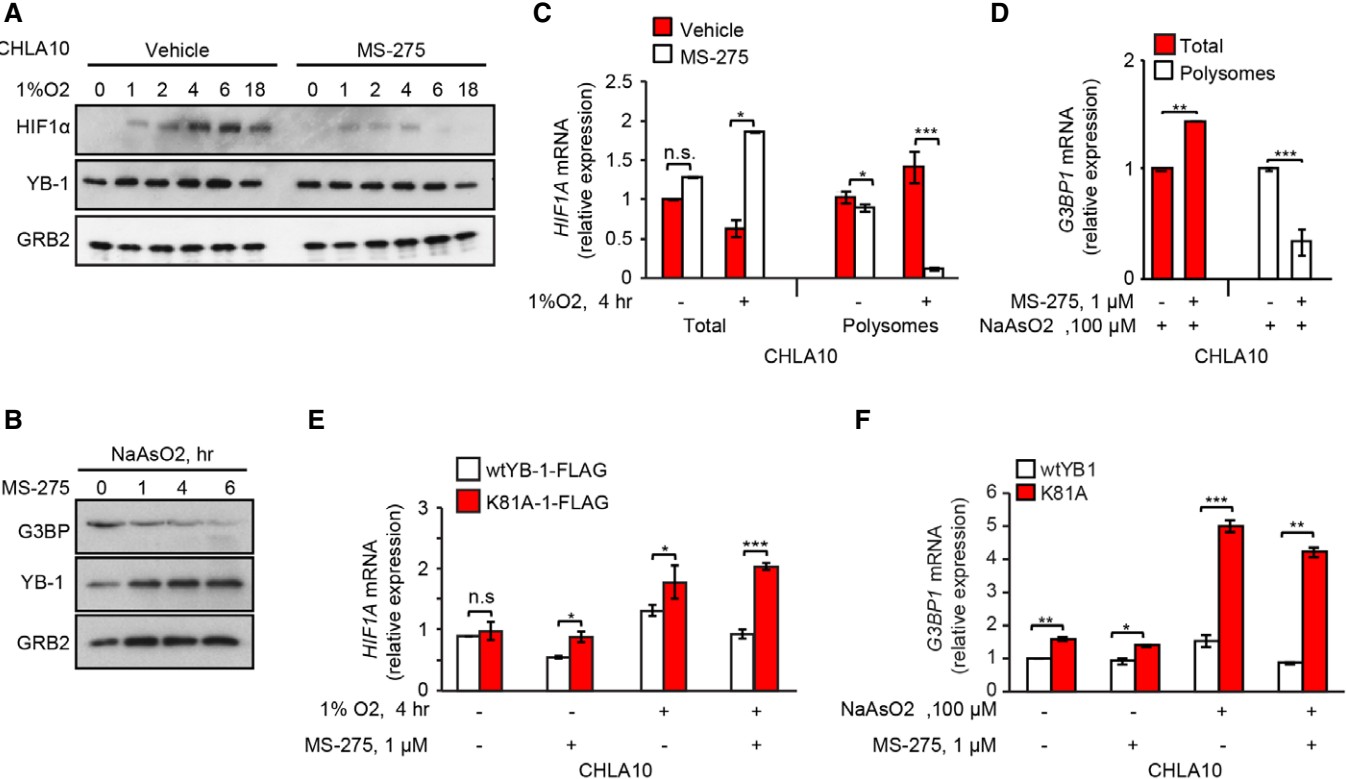

**Figure 4. MS-275-induced YB-1 acetylation inhibits YB-1 translational regulation activity.**

A  Immunoblotting showing time course analysis of HIF1α expression in response to MS-275 treatment (1 μM) in CHLA-10, incubated at 1% O₂ for the indicated times. GRB2 was used as a loading control.

B  Immunoblotting showing G3BP protein levels in U2OS cells treated with or without MS-275 (1 μM) and NaAsO₂ (100 μM) for the indicated time periods. GRB2 was used as a loading control.

C  Total and polysomal RNA isolated from CHLA-10 cells treated without (vehicle) or with MS-275 (1 μM, 24 h) and further incubated under hypoxia (1% O₂, 4 h) as indicated, was subjected to RT–PCR using *HIF1A* primers. Data were normalized against GAPDH expression. Mean values ± SEM (error bars) are shown for two independent experiments, each performed in triplicate.

D  Total RNA from CHLA-10 cells +/− MS-275 treatment (1 μM, 24 h) and NaAsO₂ (100 μM, 1 h) was subjected to polysomal fractionation. Total (red bars), and polysome-bound mRNA levels (white bars) were determined by qRT–PCR using primers for *G3BP1*. Data were normalized against *GAPDH* expression. Mean values ± SEM. (error bars) are shown for two independent experiments, each performed in triplicate.

E  *HIF1A* mRNA binding to FLAG-tagged wt or YB-1-K81A in CHLA-10 cells, subjected to the indicated treatments as described in (C), and as measured by qRT–PCR. Data were normalized for each sample against the geometric mean of *YBX1* mRNA binding and expressed as fold change ± SEM of two independent experiments, each performed in triplicate.

F  *G3BP1* mRNA binding to FLAG-tagged wt or YB-1-K81A in CHLA-10 cells +/− MS-275 (1 μM, 24 h) and NaAsO₂ (100 μM, 1 h), as measured by qRT–PCR of total RNA isolated from cell lysates. Data were normalized for each sample against the geometric mean of *YBX1* mRNA binding and expressed as fold change ± SEM of two independent experiments, each performed in triplicate.

Data information: Unpaired two-tailed Student's *t*-test;*P < 0.05; **P < 0.005; ***P < 0.0005; n.s = non-significant.
Source data are available online for this figure.

We next validated MS-275 effects on *in vivo* YB-1 acetylation, which was readily detected in CHLA-10 tumors from MS-275-treated mice but not in corresponding tumors from vehicle control mice (Fig EV4D). We also confirmed that MS-275 alters ROS levels *in vivo*. Dihydroethidium (DHE) staining, an *in vivo* marker of ROS, was significantly enhanced in CHLA-10 xenografts from MS-275-treated mice compared to controls (Fig 5A). Staining with a second oxidative stress marker, 4-hydroxynonenal (4-HNE), was also significantly higher in MS-275-treated CHLA-10 tumor xenografts compared to control tumors (Fig EV4E). Therefore, MS-275 also enhances sarcoma oxidative stress *in vivo*.

These findings were recapitulated in an EwS patient-derived (PDX) model. MS-275 treatment of mice delayed tumor growth

(Fig EV4F) and the time to humane endpoint compared with vehicle alone (Fig EV4G). PDX tumors from vehicle treated mice were locally invasive and formed lung metastases similarly to CHLA-10 cells (Fig 5B and C; see vehicle control panels). Notably, as with CHLA-10 xenografts, MS-275 dramatically reduced local invasion (Fig 5B; see bar graph on the right) and lung metastasis (Fig 5C and D), as well as the size of metastatic nodules (Fig 5E). Similar to CHLA-10 xenografts, YB-1 acetylation was readily detectable in MS-275-treated EwS PDX tumors, but not in corresponding tumors from vehicle control mice (Fig 5F). Therefore, MS-275 markedly reduces EwS invasive and metastatic capacity, and this correlates with increased YB-1 acetylation, ROS accumulation, and oxidative stress *in vivo*.

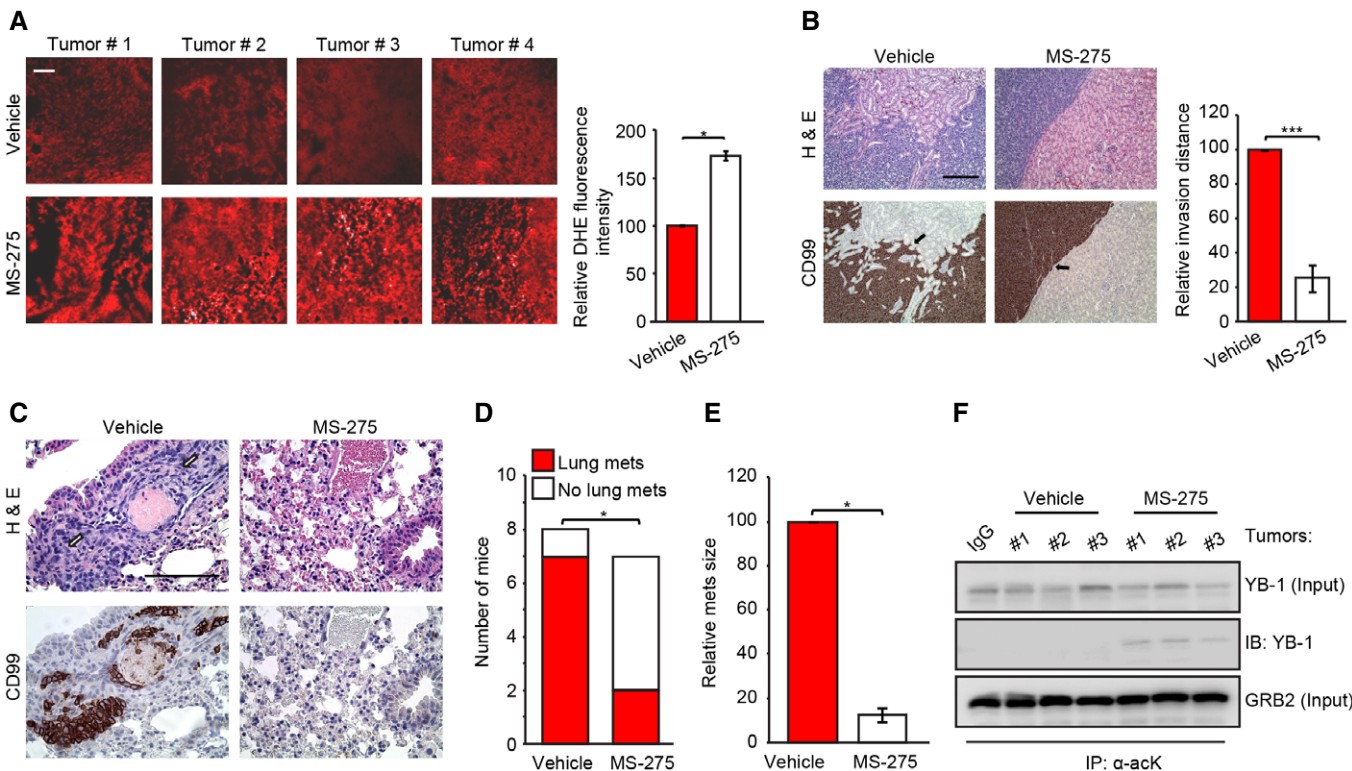

**Figure 5. MS-275 inhibits sarcoma metastasis *in vivo*.**

A   Left panel: Frozen sections of the indicated CHLA-10 tumor groups (vehicle treated and MS-275 treated) were assessed for ROS levels using dihydroethidium (DHE) staining (red). Right panel: Quantification of DHE staining in each group was assessed in 10 high-power fields per tumor (four tumors/group) using ImageJ, analyzed with unpaired two-tailed Student's *t*-test, and graphically represented. Error bars indicate SEM.

B   Left panel (Top): H&E-stained representative sections of (IC-pPDX-3) EwS PDX xenografts in NRG mice, +/− MS-275 treatment. Arrows show highly invasive growth patterns of vehicle tumor xenografts and non-invasive borders of MS-275-treated xenografts. Left panel (Bottom): immunohistochemical staining (brown) of the EwS marker CD99. Right panel: quantification of local invasion in EwS PDX xenografts in vehicle-treated and MS-275-treated groups as conducted using unpaired two-tailed Student's *t*-test. Average distances of invasion of single or tumor cell clusters at the tumor/kidney interface in 15 high-power fields per tumor (three tumors/group) were assessed, normalized to vehicle-treated group, and graphically represented. Error bars indicate SEM.

C   Top panels: H&E staining of metastatic lung lesions (arrows) in mouse xenografts in the indicated tumor groups. Bottom panels: immunohistochemical staining (brown) of the Ewing sarcoma marker CD99 in lung metastases.

D   Total number of mice bearing xenografts of the indicated PDX tumor groups that developed lung metastases, determined using a Fisher's exact test.

E   Relative mean size of lung metastases developed in NRG mice (8/group) bearing EwS PDX (IC-pPDX-3) +/− MS-275 treatment as conducted using unpaired two-tailed Student's *t*-test. Error bars indicate SEM.

F   Immunoblotting showing YB-1 acetylation *in vivo*. Tumor lysates from EwS PDX tumors +/− MS-275 treatment were subjected to immunoprecipitation with α-acK antibody as described in Fig EV4D. Acetylated YB-1 was analyzed by immunoblotting using antibodies to YB-1. IgG was used as negative antibody control, and GRB2 was used as a loading control to assess input loading.

Data information: *P < 0.05; ***P < 0.0005. All scale bars = 100 μm.
Source data are available online for this figure.

We next assessed links between MS-275 treatment and expression of YB-1 translational targets *in vivo*. By IHC, NRF2 expression in CHLA-10 xenografts was significantly reduced in MS-275-treated mice compared to controls (Fig EV5A). Notably, while NRF2 expression was enhanced at invasive tumor margins adjacent to normal kidney, compared to central areas of tumors, this was not observed in MS-275-treated tumors (Fig EV5B). These findings were confirmed in EwS PDX tumors, whereby NRF2 levels were strongly reduced in MS-275-treated compared to control tumors using IHC (Fig EV5C) and Western blotting (Fig EV5D). Therefore, MS-275 also blocks NRF2 expression *in vivo* in EwS tumors. Similarly, compared to vehicle-treated mice, CHLA-10 renal subcapsular tumors from MS-275-treated mice showed markedly reduced G3BP1 (Fig EV5E) and HIF1α protein levels (Fig EV5F). Moreover, using described methods

[32], immunofluorescence of CHLA-10 tumor sections readily demonstrated SGs in vehicle-treated tumors, but SGs were dramatically reduced in tumors from MS-275-treated group (Fig EV5G). Again, together, these findings strongly support a model whereby MS-275 increases acetylation of YB-1-K81 within the YB-1 CSD to inhibit *NFE2L2, G3BP1,* and *HIF1A* binding, in turn blocking YB-1 translational activation of these key stress-adaptive targets.

### YB-1-K81A restores YB-1 translational control of stress-adaptive targets and metastatic capacity in MS-275-treated mice

To definitively test effects of MS-275-mediated YB-1 acetylation on YB-1 pro-metastatic effects *in vivo*, we expressed control versus *wt* YB-1 and YB-1-K81A expression vectors in CHLA-10 cells and

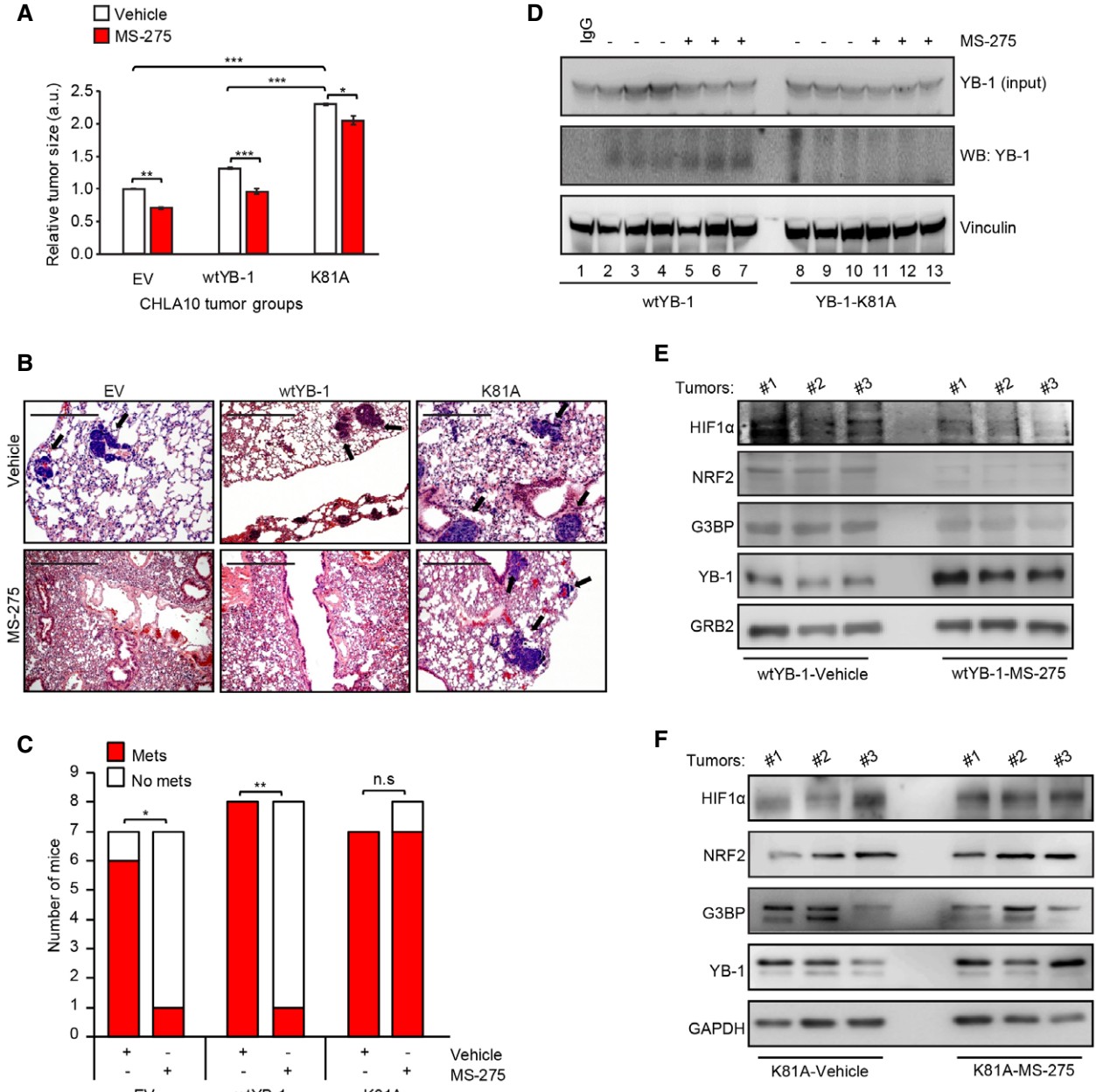

**Figure 6. YB-1-K81A rescues sarcoma cells' metastatic phenotype *in vivo*.**

A   Comparison of relative tumor sizes in CHLA-10 tumor xenografts of the indicated groups (8 mice/group), 6 weeks post-xenotransplantation +/− MS-275 treatment was conducted using unpaired two-tailed Student's *t*-test. Error bars indicate SEM.

B   H&E staining of metastatic lung lesions (arrows) in mice with renal subcapsular tumor xenografts of CHLA-10 cells expressing empty vector (EV), wtYB-1, or YB-1K81A, and treated with vehicle or MS-275 as described in Fig 5B. Scale bars = 100 μm.

C   Total number of mice bearing xenografts of the indicated CHLA-10 tumor groups that developed lung metastases, determined using a Fisher's exact test.

D   Immunoblotting showing YB-1 acetylation *in vivo*. Tumor lysates from the indicated tumor groups, +/− MS-275 treatment were subjected to immunoprecipitation with α-acK antibody as described in Fig EV4D. Acetylated YB-1 was analyzed by immunoblotting using antibodies to YB-1. IgG was used as negative antibody control, and Vinculin was used as a loading control to assess input loading.

E   Immunoblot showing HIF1α, NRF2, G3BP, and YB-1 expression in tumor lysates of CHLA-10 cells expressing wtYB-1, +/− MS-275 treatment from three independent mouse tumors (vehicle or MS-275 #1-3) per group. GRB2 was used as a loading control. Note that HIF1α, NRF2, and G3BP are all downregulated in tumor lysates from MS-275 treated mice.

F   Immunoblot showing HIF1α, NRF2, G3BP, and YB-1 expression in tumor lysates of CHLA-10 cells expressing YB-1-K81A, +/− MS-275 treatment from three independent mouse tumors (vehicle or MS-275 #1-3) per group. GAPDH was used as a loading control. Note the equal expression of HIF1α, NRF2, and G3BP in tumor lysates +/− MS-275 treatment.

Data information: *$P < 0.05$; **$P < 0.005$; n.s = non-significant.
Source data are available online for this figure.

implanted them under the renal capsules of immunocompromised mice as above. Mice were treated with vehicle or MS-275 and monitored for tumor growth and metastatic spread as in Fig 6A. While *wt* YB-1 slightly increased primary tumor growth compared to vector alone cells, YB-1-K81A tumors were significantly larger than those from the other mouse cohorts (Fig 6A); MS-275 moderately but significantly reduced primary tumor sizes in all cohorts compared to vehicle treatment, albeit less in the YB-1-K81A cohort (Fig 6A). Strikingly, however, while MS-275 dramatically reduced metastatic growth of vector alone and *wt* YB-1-expressing tumors to lungs (Fig 6B and C), similar to the results of Figs EV4A and 5C–E, metastatic capacity was almost completely rescued in YB-1-K81A-expressing cells even with identical MS-275 treatment. Moreover, while MS-275 increased acetylated YB-1 in primary tumors from the *wt* YB-1 overexpression cohorts (Fig 6D; compare lanes 2–4 to 5–7), we failed to detect YB-1 acetylation in the YB-1-K81A drug-treated primary tumors (Fig 6D; lanes 8–13). This correlated strongly with expression of YB-1 translationally controlled proteins, as G3BP1,

HIF1α, and NRF2 were all reduced by MS-275 in *wt* YB-1 tumors (Fig 6E), but expression of each protein was retained in YB-1-K81A drug-treated primary tumors (Fig 6F). This unequivocally demonstrates that enhanced acetylation of YB-1 K81 by MS-275 blocks YB-1-mediated pro-metastatic activity in sarcoma cells.

Finally, we analyzed publicly available EwS and OS gene expression datasets and found moderate correlations between *HIF1A*, *G3BP1*, and *NFE2L2* transcripts in EwS and OS (Fig 7A and B); moreover, *G3BP1* levels correlated significantly with poor outcome in both diseases (Fig 8A). We then tested an available OS tissue microarray by IHC, where we observed significant correlations between G3BP1, HIF1α, NRF2, and YB-1 staining (Fig 8B and C, top panels). Moreover, co-expression of G3BP1, HIF1α, NRF2, and YB-1 staining was strongly correlated with advanced disease in both EwS and OS (Fig 8B and C, lower panels). Together, these findings point to previously unrecognized correlations between different YB-1 translationally controlled targets in high-risk childhood bone sarcomas. We speculate that enhanced YB-1 acetylation at K81 via

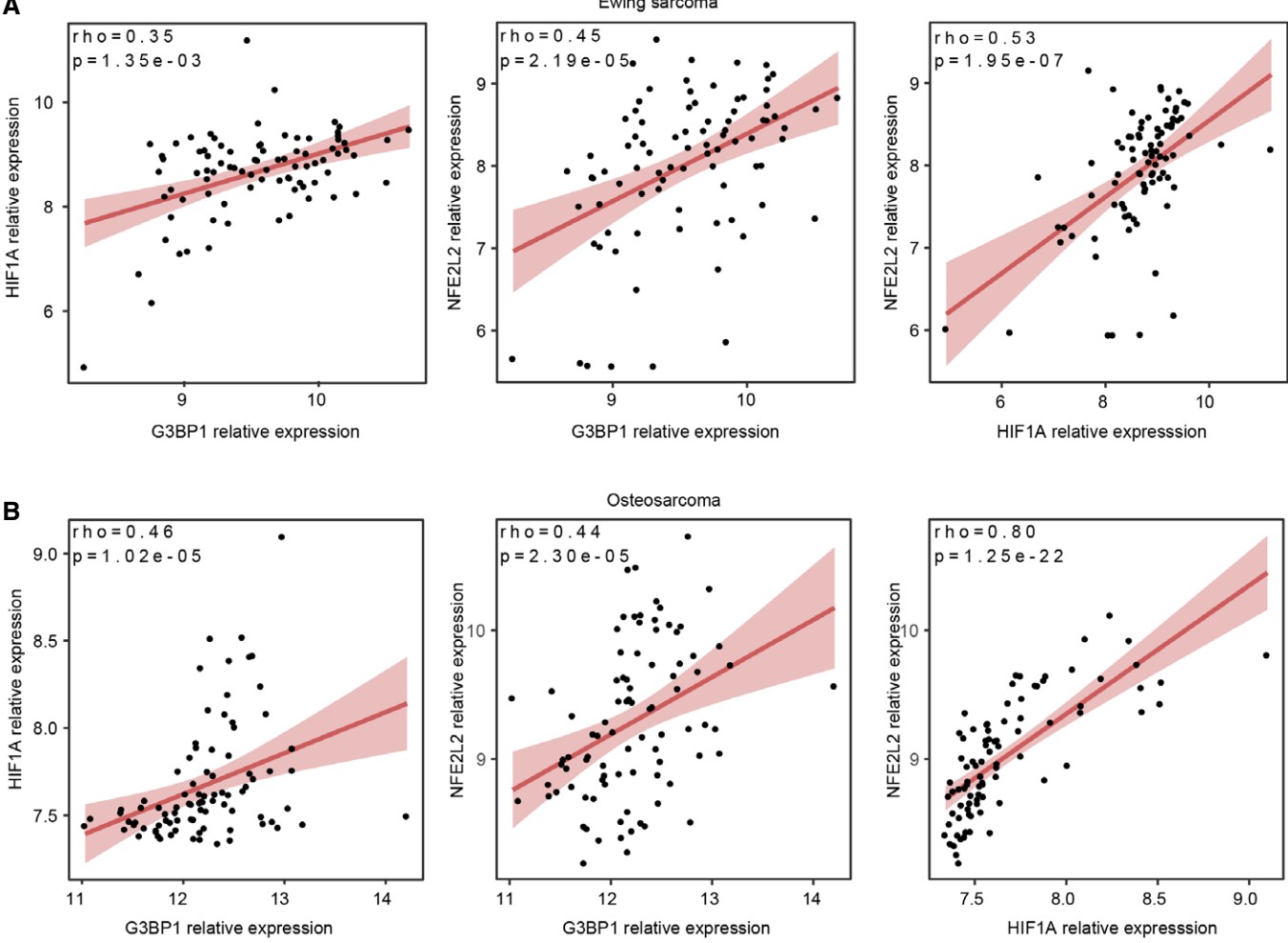

**Figure 7.  Correlation between YB-1 expression and HIF1α, NRF2, and G3BP1 levels in EwS and OS.**

A, B    Scatterplots showing correlations between *G3BP1-HIF1A* (left panel), *G3BP1-NFE2L2* (middle panel), and *HIF1A-NFE2L2* (right panel) mRNA expression in Ewing sarcoma (A, top panels) and osteosarcoma (B, bottom panels), quantified with Spearman's correlation, using publicly available cohorts of EwS (GSE63157) (A; top panel), and OS (GSE42352) (B; bottom panel), respectively. rho: Spearman's correlation coefficient.

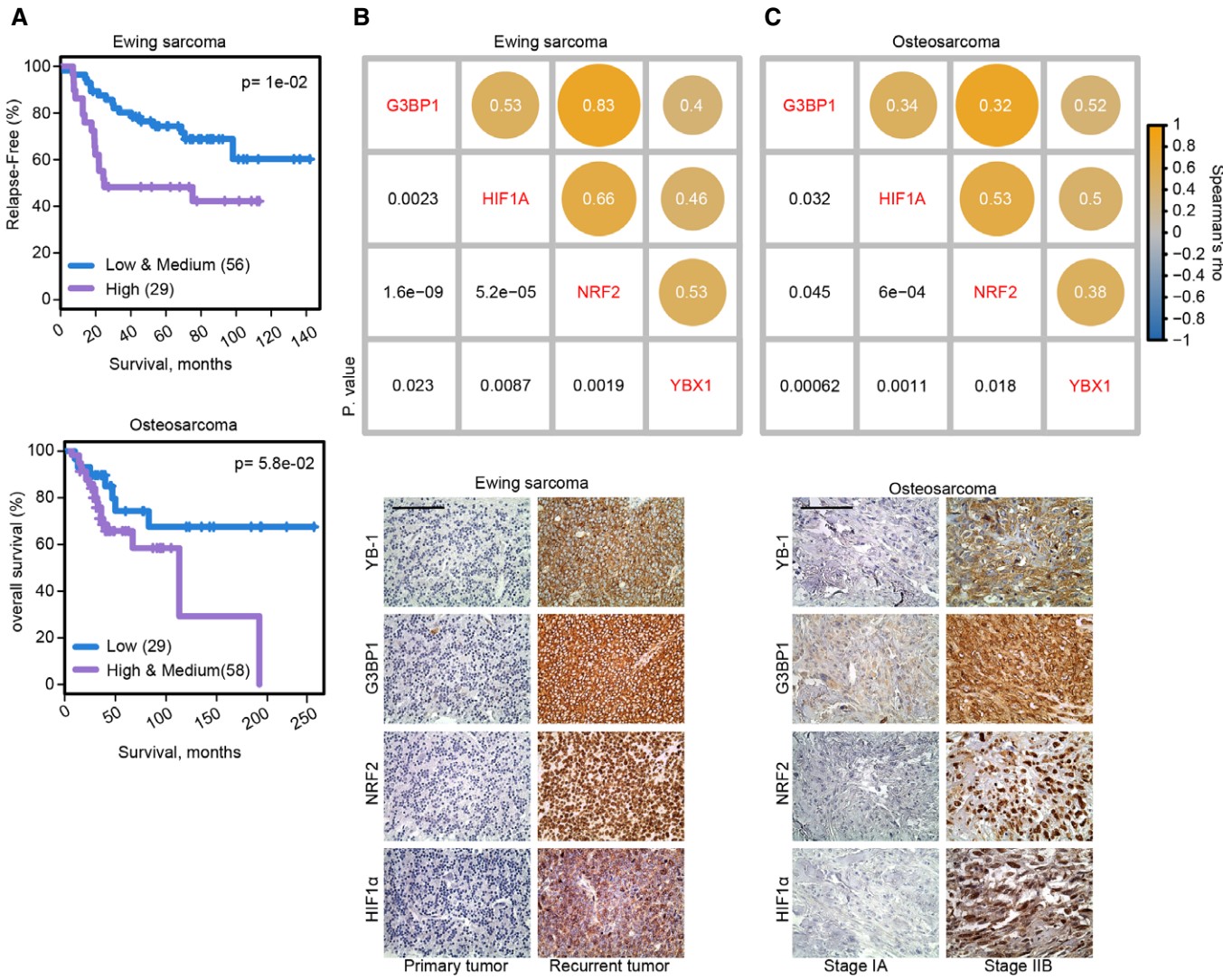

**Figure 8. Co-expression of YB-1 targets in clinical samples.**

A  Kaplan–Meier plots, with *P*-values computed with a log-rank test, showing relapse free survival in Ewing sarcoma (top panel) and overall survival in osteosarcoma patients (bottom panel) based on *G3BP1* mRNA expression. Clinical data were obtained from the GSE63157 publicly available gene expression dataset for EwS and from http://r2.amc.nl for osteosarcoma.

B, C  Top panels: Immunohistochemistry (IHC) was conducted on serial sections of EwS (B) and OS (C) TMAs using antibodies against YB-1, G3BP1, NRF2, and HIF1α. Spearman's rank correlation coefficient (rho; brown color), computed on H-scores (staining intensity × percentage) for the different markers, along with relative *P*-values in black color in the lower left quadrants are shown. Bottom panels: representative IHC images for YB-1, G3BP1, NRF2, and HIF1α as indicated, on serial histological sections of EwS (bottom panel B; comparing primary to recurrent paired specimens) and OS (bottom panel C; comparing Stage IA to Stage IIB paired specimens). Scale bars = 100 μm.

Source data are available online for this figure.

MS-275 prevents binding and translational activation by YB-1 of multiple stress-adaptive mRNAs, including *G3BP1*, *NFE2L2*, and *HIF1A*, and likely others, although further studies are necessary to evaluate the full mRNA spectrum affected by this process.

## Discussion

In EwS and OS, the single most powerful predictor of poor outcome is the presence of metastatic disease. However, stresses arising during the metastatic cascade, such as ROS-induced oxidative stress, can potentially cull tumor cells prior to metastatic colonization, rendering metastasis a highly inefficient process. Here, we show that the class I HDAC inhibitor, MS-275, in clinical trials for advanced solid tumors [51,52], induces oxidative stress in EwS and OS cells and dramatically reduces their metastasis *in vivo*. Class I HDAC inhibitors are known to prevent histone deacetylation, leading to chromatin remodeling and transcription-based anti-neoplastic activity [41], which is the basis for their broad clinical application [51,52]. HDAC inhibitors are also reported to enhance ROS levels in melanomas [53] and other tumors [54–57] by unknown mechanisms, but other studies report

modest or no effects alone [58], or even reduced ROS after long-term HDAC inhibitor treatment [59]. However, we find that MS-275 enhances ROS indirectly in sarcoma cells by non-transcriptionally inhibiting YB-1-mediated translation of *NFE2L2* encoding the antioxidant master regulator, NRF2. Translation and synthesis of NRF2, as well as of other known YB-1 targets, G3BP1 and HIF1α, were blocked by MS-275 in sarcoma cells, due to enhanced acetylation of lysine 81 within the YB-1 RNA-binding domain and consequent loss of YB-1 translational activation of these stress factors. However, each protein was rescued by ectopic expression of an MS-275-resistant YB-1-K81A mutant. Moreover, YB-1-K81A rescued EwS metastasis to lungs, which correlated with re-expression of NRF2, G3BP1, and HIF1α in tumor tissues. We postulate that collectively, this decreases the fitness of sarcoma cells to metastasize. Therefore, inhibiting YB-1 translational activity has widespread effects on its pro-metastatic activities, providing a compelling rationale for its targeting in bone sarcomas. HDAC inhibitors are known to enhance ROS levels, such as in melanoma cells [53] and other tumors [54–57]. MS-275, which preferentially targets HDAC1 and HDAC3, is reported to induce acute ROS in human leukemia cells [60]. While HDACs may affect histone acetylation within hours [61], resulting phenotypic changes in target cells through changes in gene expression typically require several cell cycles (i.e., ~24 h) [62]. We observed that MS-275 increases YB-1 acetylation as early ~30 min after initiation of treatment. Although not strictly ruling out transcriptional effects of MS-275, these findings argue for a more direct action of this agent on YB-1 functions through enhanced K81 acetylation. Possible mechanisms whereby this could affect YB-1 translational control activity include direct steric hindrance due to K81 acetylation, or recruitment of acetyl-lysine interacting proteins to acetylated K81, either of which could block YB-1 binding to its target mRNAs. Notably, mutation of YB-1-K81 (K81E) has been reported in human tumors (http://cancer.sanger.ac.uk/cosmic/gene/analysis?ln = YBX1), although the functional significance of this finding remains to be established. Of interest, YB-1 was recently shown to form nucleoprotein filaments with bound mRNAs in the cytoplasm, thus potentially increasing translational efficiency of its target mRNAs by converting them into extended linear forms [63]. It will be of interest to determine how YB-1 acetylation by MS-275 affects this function.

Our findings highlight *NFE2LE* as a new translational target of YB-1. Links between ROS and cancer progression are well documented [64]. However, elevated ROS also impedes cell fitness and therefore must be kept under tight control to prevent ROS-induced toxicity [6]. NRF2-activating or KEAP1-inactivating mutations [65], or oncogene-mediated transcriptional induction of NRF2 [7], contribute to NRF2 activation in diverse tumor types. Loss of NRF2 in pancreatic carcinoma cells oxidizes components of the translational machinery, leading to impaired mRNA translation and reduced proliferation [17]. YB-1 translational activation of *NFE2L2* provides an important additional level of control of NRF2 expression under stress, at least in EwS and OS. NRF2 could potentially reduce oxidative stress at one or more critical steps to facilitate tumor cell fitness within the metastatic cascade, consistent with oxidative stress preventing melanoma metastasis [4,5]. Decreased NRF2 activity reduces trophoblast invasion in placenta [66], and KEAP1 overexpression suppresses migration of lung adenocarcinoma cells [67].

Other steps in the metastatic cascade, such as survival in the circulation, may also require NRF2 activation to block anti-metastatic ROS accumulation.

Exactly how YB-1 confers fitness for sarcoma cells to survive and colonize lungs or other distant sites as part of the metastatic cascade remains to be determined. We speculate that translational activation by YB-1 of multiple, rather than single, stress-adaptive mRNAs, such as *NFE2L2*, *G3BP1*, and *HIF1A*, but likely others, functions collectively to facilitate metastasis. This model would provide a broad level of cell plasticity to sarcoma cells, allowing them to respond translationally through up-regulation of context-specific sets of survival factors as needed under different stress conditions, rather than by relying only on an individual effector. Translational regulation would also provide an acute means for tumor cells to respond to rapidly changing microenvironments. Indeed, stress-induced nuclear NRF2 accumulation is largely due to *de novo* protein synthesis, rather than from translocation from a pre-existing pool [9], pointing to translational control as a key component of the NRF2 response under oxidative stress.

In summary, these studies further highlight YB-1 as a potent metastatic driver in high-risk childhood bone sarcomas by enhancing tumor cell fitness during the metastatic cascade, including through an elevated NRF2-mediated antioxidant response. Targeting YB-1 translational control functions with HDAC inhibitors to limit its ability to bind key stress-adaptive transcripts may provide a more tractable strategy than, for example, inhibiting YB-1 expression or its binding to protein interactors, given its highly disordered topography [18]. Using molecules such as MS-275, alone or in combination with other agents, represents an exciting strategy for therapeutic intervention in sarcomas, particularly for reducing their metastatic capacity.

# Materials and Methods

### Clinical specimens

All studies involving IHC on human tissue samples were approved by the University of British Columbia Research Ethics Board (UBC REB Number H14-01553).

### Animal studies

We used murine renal subcapsular implantation model, which is a well established model for studying human tumor xenografts [49], and which gives rise to macroscopic, quantifiable high-volume metastases of sarcoma tumor cell lines [50,68]. All studies are conducted under the UBC animal care certificate # A10-0200. In brief, xenograft cell blocks for implantation were prepared using $1 \times 10^6$ CHLA-10 EwS cells. Then, cells were implanted under the renal capsules of immunocompromised mice; 6- to 8-week-old NRG male mice from Animal Resource Centre BC Cancer Research Center (8–10 per condition), animals were maintained according to UBC Animal Care Committee (ACC) regulations. Three weeks post-inoculation, MS-275 was administered orally at 20 mg/kg, 5 days/week for 3 weeks, resulting in two experimental groups, namely vehicle treated (0.05 N HCl and 0.1% Tween-80 in saline solution) and MS-275-treated groups. Mice were then euthanized, and tumors were collected. Animal studies involving CHLA-10 cells

expressing empty vector (EV), wtYB-1, or YB-1-K81A followed the same protocol as described above. IC-pPDX-3 model was established from a fresh Ewing sarcoma at the time of diagnostic. The tumor was grafted in the subscapular fat pad (without any dissociation) of CB17 SCID ♀ (CB17/Icr-Prkdcscid/IcrIcoCrl) mice (4/ group). Successive passages of undissociated tumors fragments were performed in the same location. The tumors showed nice and homogenous growth pattern across passages; 1.5 months to reach tumor at ethical size: 2 cm$^3$. This PDX carries the classical EWSR1-FLI1 (type 1) translocation and displays a strong CD99 membranous staining by IHC. In the current study, EwS PDX models were established by renal subcapsular implantation of tumor fragments in NRG mice (8/group). Three weeks post-implantation, tumors had reached a size ~100–150 mm$^3$, and mice were randomized into two groups and MS-275 treatment was initiated. MS-275 was administered orally at 20 mg/kg, 5 days/week versus vehicle alone. When reaching a humane endpoint, mice were euthanized and tumors along with other organs were collected. Histological evaluation of xenografts was performed on 4-µm-thick sections stained with hematoxylin and eosin and examined using Axioplan2 fluorescence microscope (Zeiss). Further IHC on tissue sections was conducted on a Ventana Discover XT system using antibodies against the G3BP1 at a dilution of 1:1,500, CD99 at 1:400, and anti-NRF2 at 1:150 with signal stain booster, and anti-4-HNE, a marker of oxidative stress [69], at 1:50. Assessment of the depth of tumor cell invasion from H&E-stained tissue sections and staining intensity were conducted as previously described [19]. Quantitative analysis of IHC samples was conducted using ImageJ software, the color deconvolution plug-in implementing stain separation as previously described [19].

## Cell lines

The CHLA-10 EWS cell line was kindly provided by Dr. Patrick Reynolds (Texas Tech University). U2OS cell lines were obtained from the American Type Culture Collection (ATCC; ATCC HTB-96). All cell lines were maintained in a 5% CO$_2$ incubator at 37°C and were verified to be free of mycoplasma contamination. For normoxic conditions, cell lines were incubated at 37°C in a humidified 21% O$_2$, 5% CO$_2$ atmosphere. Hypoxia (1% O$_2$, 5% CO$_2$, 95% N$_2$) was achieved using a hypoxic incubation chamber (COY, Laboratory Products, Inc., Grass Lake, MI). The key resources and reagents used are listed in Appendix Table S2.

## Cell-based screening of oxidative stress sensitizers in sarcoma cells

U2OS cells were seeded at 6,000/well in 96-well plate at day 1. At day 2, cells were exposed to the Cayman Cat#11076 epigenetic drug library, including 92 chemicals directed against known epigenetic modifiers, including methyltransferases, demethylases, histone acetyltransferases, HDACs, and acetylated lysine reader proteins at 1 µM for 24 h. NaAsO$_2$ was then added to cells to a final concentration of 100 µM for 1 h. Then, cells were washed, and fresh medium was added along with fresh compounds' treatments and NucView-488 dye, which binds to activated caspase-3/7 for monitoring cell apoptosis by Incucyte live cell monitor. The apoptotic counts at 12 h after NaAsO$_2$ treatment were used for calculation of z-score

using the formulation: $Z = (X_i - X_{dmso})$/standard deviation, where $X_i$ and $X_{dmso}$ stand for the apoptotic counts of individual treatment and DMSO treatment, respectively. Average z-scores from three independent experiments were then calculated and plotted. $Z = 1$ was used as a cutoff.

## Stress granule detection

Stress granule (SG) detection and quantification were conducted as previously described [32], with at least 3 SGs per cell were required to consider as positive scoring [70].

## SILAC labeling, acetylated protein enrichment, and mass spectrometry analysis

U2OS cells were grown in custom DMEM depleted of lysine and arginine supplemented with 10% dialyzed fetal bovine serum (dFBS) and 1× non-essential amino acids. For "light" cultures (mock-treated, ± NaAsO$_2$), the medium was supplemented with 100 mg/l of [$^{14}$N$_2$ $^{12}$C$_6$]-Lys and 100 mg/l of [$^{14}$N$_4$ $^{12}$C$_6$]-Arg. For "heavy" cultures (+MS-275, ± NaAsO$_2$), the medium was supplemented with 100 mg/l of [$^{15}$N$_2$, $^{13}$C$_6$]-Lys and 100 mg/l of [$^{15}$N$_4$, $^{13}$C$_6$]-Arg. Then, cells were harvested, washed twice with cold phosphate-buffered saline (PBS), and lysed in 500 µl NP-40 lysis buffer containing 1× protease inhibitor and 5 mM DTT followed by centrifugation at 10,000 g for 5 min. Protein concentrations of the pre-cleared lysates were detected using the Quick-start Bradford assay, and equal amounts (1:1) of heavy and light labeled protein were mixed and then incubated with 50 µl anti-acetyl-lysine antibody conjugated to agarose (50 µl/IP) overnight (O/N) at 4°C on a rotating wheel. Beads were then washed: four times with NP-40 buffer and two times with dH$_2$O. Then, protein elution was conducted using 2× LDS Sample buffer non-reducing (lithium dodecyl sulfate sample loading buffer) (50 µl/IP). For each sample, 800 µl of lysis buffer (100 mM HEPES pH 8) (CAT#H3375, Sigma), 4M guanidine hydrochloride (Cat # G4505, Sigma), 10 mM TCEP (CAT#C4706, Sigma), 40 mM CAA (CAT#C0267, Sigma), and 1× cOmplete protease inhibitor—EDTA free (Cat # 4693159001, Sigma) was added. Lysis mixtures were passed through a 21-gauge needle mounted to a 1 ml syringe (BD Biosciences) a total of five times, centrifuged at 20,000 g for 2 min, and the supernatant recovered. Resultant lysates were heated at 90°C for 15 min and chilled to room temperature for a further 15 min. For digestion, lysates were diluted 1:10 in 0.2 M HEPES pH 8 containing a 1:50 (µg:µg) enzyme to protein amount of trypsin/rLysC mix (Promega, Cat #V5071). Mixtures were incubated for 14 h at 37°C in a thermomixer with shaking at 200 g. Resultant digests were acidified to 1% trifluoroacetic acid, precipitate pelleted with 2 min at 20,000 g, and filtered through an Amicon Ultra 10 kDa unit (12,000 g for 10 min). Filtered digests were cleaned up with a SepPak prior to MS analysis. Peptide analysis was carried out on an Orbitrap Fusion Tribrid MS platform (Thermo Scientific). Samples were introduced using an Easy-nLC 1000 system (Thermo Scientific). Columns used for trapping and separations were packed in-house. Trapping columns were packed in 100 µm internal diameter capillaries to a length of 25 mm with C18 beads (Reprosil-Pur, Dr. Maisch, 3 µm particle size). Trapping was carried out for a total volume of 10 µl at a pressure of 400 bar. After trapping, gradient elution of peptides was performed

on a C18 (Reprosil-Pur, Dr. Maisch, 1.9 μm particle size) column packed in-house to a length of 15 cm in 100 μm internal diameter capillaries with a laser-pulled electrospray tip and heated to 45°C using AgileSLEEVE column ovens (Analytical Sales & Service). Elution was performed with a gradient of mobile phase A (water and 0.1% formic acid) to 25% B (acetonitrile and 0.1% formic acid) over 32 min, and to 40% B over 4 min, with final elution (80% B) and equilibration (5% B) using a further 4 min at a flow rate of 400 nl/min. Data acquisition on the Orbitrap Fusion (control software version 2.1.1565.20) was carried out using a data-dependent method. Survey scans covering the mass range of 350–1,500 were acquired at a resolution of 120,000 (at $m/z$ 200), with quadrupole isolation enabled, an S-Lens RF Level of 60%, a maximum fill time of 50 ms, and an automatic gain control (AGC) target value of 5e5. For MS2 scan triggering, monoisotopic precursor selection was enabled, charge state filtering was limited to 2–4, an intensity threshold of 5e3 was employed, and dynamic exclusion of previously selected masses was enabled for 60 s with a tolerance of 20 ppm. MS2 scans were acquired in the ion trap in Rapid mode after HCD fragmentation with a maximum fill time of 150 ms, quadrupole isolation, an isolation window of 1 $m/z$, collision energy of 35%, activation Q of 0.25, injection for all available parallelizable time turned OFF, and an AGC target value of 4e3. The total allowable cycle time was set to 4 s. MS1 scans were acquired in profile mode, and MS2 in centroid format. All MS data were analyzed with MaxQuant (development version 1.2.7.1). All SILAC pairs were quantified, and MS/MS spectra were searched against the human UniProt FASTA database (February 2012 release) to identify corresponding proteins. Cysteine carbamidomethylation was searched as a fixed modification, whereas methionine oxidation, N-acetyl protein, and acetylation of lysine were chosen as variable modifications. The false discovery rate (FDR) was fixed to a threshold of 1% FDR, and all peptide identifications were filtered for length and mass error. For quantification purposes, in order to reduce dubious identifications, only proteins with 2 or more "unique peptides" were included in the analysis. Heatmap of the H/L SILAC ratios of histone proteins and stress granules-related proteins was plotted in R software with "pheatmap" package [71].

For detection of YB-1 acetylated residues, U2OS cells growing in light (mock-treated, +NaAsO₂) or heavy SILAC media (+MS-275, +NaAsO₂) as described above were harvested, washed in PBS, and lysed in 500 μl NP-40 lysis buffer containing 1× protease inhibitor (CAT# 11836170001, Sigma) and 5 mM DTT followed by centrifugation at 10,000 $g$ for 5 min. Protein concentrations of the pre-cleared lysates were detected using Quick-start Bradford assay. In parallel, anti-YBX1 antibody (RN015P, MBL International Corporation) at 5 μg/250 μl of cell extract was incubated with protein A/G magnetic beads (50 μl/reaction) suspended in lysis buffer for 3 h on a rotating wheel at room temperature. Subsequent to washing the beads, 500 μl of pre-cleared cell lysate was incubated with the beads overnight on a rotating wheel at 4°C. Beads were then washed four times, and proteins were eluted using 2× LDS Sample buffer nonreducing (50 μl/IP) at 95°C for 10 min. Eluted samples were run on NuPAGE 4-12% Bis-Tris Gel (Cat # NP0336BOX, Invitrogen), which was subsequently stained using Coomassie stain. The gel was scanned, and the YB-1 band was cut, digested, and processed for MS analysis as described previously [72]. Peptide analysis on a Orbitrap Fusion Tribrid MS platform (Thermo Scientific) is

described above. Cysteine carbamidomethylation was searched as a fixed modification, whereas methionine oxidation and acetylation of lysine were chosen as variable modifications.

### *In vitro* ROS measurements

Intracellular generation of reactive oxygen species was detected using the membrane permeable, fluorogenic probes CM-H2DCFDA, CellROX, or DHE (Invitrogen).

CM-H2DCFDA was used as a general oxidative stress indicator. Briefly, tumor cells were seeded in 6-well plates and treated with selected compounds (+/MS-275, +/− NaAsO₂). CM-H2DCFDA was added to the cultured cells (60–80% confluence) at a final concentration of 10 μM for 30 min. Subsequently, cells were trypsinized and cell pellet collected. Cells were washed 2–3× in 1× PBS and centrifuged at 1,000 $g$ for 3–4 min at 4°C after each wash. Then, cell pellets were suspended in 300 μl PBS containing Hoechst 33342 and transferred to pre-labeled FACS tubes (Cat # 352052, BD/VWR) for analysis of CM-H2DCFDA fluorescence using the LSRFortessa multicolor analyzer. The percentage of CM-H2DCFDA-positive stained cells were recorded from a total of 10,000 cells using FL1 detector at 488 nm and 530-30-A filter. The obtained data were processed and analyzed using the FlowJo 7.1.6. software (OR, USA). Alternatively, cells were seeded in 96-well plate (50,000/well) and treated as previously indicated. The CM-H2DCFDA fluorescence was measured using plate reader (SpectraMax i3x) (excitation/emission at 485/535).

CellROX: U2OS cells were seeded in 96-well plate (50,000/well) and treated as previously indicated. Then, cells were stained with 5 μM CellROX™ Deep Red reagent (red) for 50 min at 37°C. Then, cells were washed three times with PBS and the red fluorescence was measured using a plate reader (SpectraMax i3x) (excitation/emission at 640/665).

DHE (Dihydroethidium) stain: In brief, 20-μm-thick sections of frozen tumors were incubated with 10 μM DHE dissolved in methanol in dark for 30 min in room temperature (RT). Then, sections were adhered on covered slips and mounted with fluorescent mounting medium (Vectorshield mounting medium for fluorescence with DAPI, Cedarlane) and immediately imaged using an epifluorescent microscope (Axio Observer Z1; Carl Zeiss; Excitation 540 nm and emission 605 nm) using a 20× objective lens. All images were captured using identical exposure times, and signal intensities were analyzed using ImageJ software.

### ROS measurement in suspension cells (3D cultures)

Intracellular ROS levels of U2OS cells were determined using CM-H2DCFDA. In brief, U2OS cells were pre-treated with either 5 mM N-acetylcysteine (NAC) 3 h, for 48 h prior to MS-275 treatment. Following 18 h of MS-275 along with the corresponding antioxidants, cells were transferred to ultra-low attachment surface plates and cultured for additional 6–18 h with continuation of MS-275 and antioxidant treatments. Cells were then incubated with CM-H2DCFDA (10 μM) for 30 min at 37°C before fluorescence was measured. CM-H2DCFDA-labeled cells were then collected, trypsinized and washed twice by spinning down, and suspended in DPBS at $1 \times 10^6$ cells/ml and transferred to black 96-well plates (50,000 cell/well). Spinning down of the plate for 1 min was conducted

prior to detecting CM-H2DCFDA fluorescence signals using plate reader (SpectraMax i3x) (excitation/emission at 485/535).

## GSH/GSSG assays

GSH/GSSG ratio was determined using GSH/GSSG-Glo™ Assay. In brief, cells were seeded onto 96-well plates at 10,000 cells per well and were allowed to attach overnight. Then, cells were treated with vehicle or MS-275 (1 µM 24 h), +/− NaAsO₂ (500 µM, 1 h). The levels of reduced and oxidized glutathione were measured according to the manufacturer's protocol.

## Sucrose gradient centrifugation and polysomal fractionation

Polysomal fractionation to categorize translationally active transcripts was performed using sucrose gradient centrifugation as previously described [19,21]. In brief, U2OS cells were first lysed using Nonidet P-40 lysis buffer. Then, nuclei and cell debris were cleared by centrifugation at 12,000 $g$ for 20 min at 4°C. Supernatants were loaded onto 30% (w/v) sucrose gradients and centrifuged at 100,000 $g$ for 15–20 min (Beckman coulter SW41 Ti rotor) at 4°C. Pellets were dissolved in NP-40 lysis buffer. This was followed by RNA extraction and cDNA synthesis and real-time PCR using primers as indicated.

## L-azidohomoalanine (AHA) labeling to identify newly synthesized proteins

L-azidohomoalanine (AHA) labeling of newly synthesized proteins was conducted as previously described [47]. In brief, U2OS cells or CHLA10-expressing FLAG-tagged wtYB-1 or YB-1K-to-A mutants were further treated with MS-275 (1 µM, 2 h). Then, cells were incubated for 45 min in methionine-free medium, with continuation of MS-275, to deplete endogenous methionine, followed by incubation with 50 µM AHA for 1 h along with NaAsO₂ (100 µM) and MS-275 (1 µM) for 1 h. For hypoxia conditions, cells were incubated for 45 min in methionine free medium, followed by incubation with 50 µM AHA along with MS-275 (1 µM) for 4 h. Cells were then washed twice with phosphate-buffered saline, scraped in NP-40 lysis buffer, and then sonicated for 10 s and centrifuged to pellet cellular debris. Protein quantification was performed using Bradford protein assays, and 500 µg of lysates was subjected to Click reactions according to the manufacturer's instructions using Click-iT® Protein Reaction Buffer Kit. Total proteins from Click reactions were precipitated with methanol/chloroform. The resolubilized biotin-tagged proteins were then incubated with 50 µl of Streptavidin-coupled magnetic beads overnight at 4°C followed by washing. Resin suspensions were incubated in 50 µl of 2× loading blue for 10 min at 95°C to separate out the tagged proteins from beads. The immunoprecipitated proteins were subjected to SDS–PAGE and immunoblotting analysis using antibodies directed against proteins of interest (NRF2, HIF1α, G3BP, and GRB2).

## RNA isolation & RT–PCR

Total RNA was purified with the RNeasy Plus Universal Kit, according to the manufacturer's guidelines. RNA yield and integrity were assessed using a NanoDrop Spectrophotometer (ND-1000). Reverse transcription of cDNA was conducted using a High-Capacity cDNA Reverse Transcription kit. Real-time polymerase chain reaction (RT–PCR) was performed in triplicate using Fast SYBR Green Master Mix and the indicated primers (shown in Appendix Table S2) on a QuantStudio™ 6 Flex Real-Time PCR system (Life Technologies). Data from two independent experiments, each performed in triplicate, were normalized against endogenous GAPDH, unless otherwise indicated, and presented as fold change relative to control sample.

## RNA immunoprecipitation

To probe for direct and preferential binding of the YB-1-K81A mutant to *HIF1A*, *G3BP1,* and *NFE2L2* mRNAs for translation activation, we expressed FLAG-tagged wtYB-1 or FLAG-tagged YB-1 K81A mutant, prepared at GenScript, in U2OS cells. Then, total cell lysates were subjected to immunoprecipitation (IP) using anti-FLAG® M2 Magnetic Beads. The beads were resuspended in TRIzol RNA extraction reagent (1 ml). Full protocol could be found here: http://www.abcam.com/epigenetics/rna-immunoprecipitation-rip-protocol. Isolated RNA was quantified, and cDNA was synthesized for subsequent quantitative RT–PCR analysis using primers listed above. Data were normalized against the geometric mean of YBX1 mRNA binding and expressed as the geometric mean fold change ± SEM of three replicates for $n = 2$ independent experiments.

## Immunohistochemistry (IHC) of tissue microarrays (TMAs)

TMAs consisting of formalin-fixed, paraffin-embedded human tissues from primary tumors of Ewing sarcoma (31 cases, provided by Children Oncology Group), and osteosarcoma (40 cases, OS804b, Biomax: https://www.biomax.us/tissue-arrays/Bone_and_Cartilage/OS804b), were stained for YB-1, HIF1α, G3BP1, and NRF2. The percentages of cells positively staining for HIF1α, G3BP1, and NRF2, as well as staining intensity, were evaluated. For the percentage of positive cells, the staining patterns of tumor cells were divided into two categories: (i) positive (whether cytoplasmic, nuclear, or nucleo-cytoplasmic) and (ii) negative (with no detectable staining). For staining intensity, we used a 4-point scale (0–3+) as previously described [73]. Cases were scored by board-certified pathologists (P.H.S and A.M.E). IHC H-score correlation analysis was performed with R software, package corrplot [74].

## Statistical analyses

All statistical analyses were conducted using Student's two-tailed $t$-test for analysis, unless otherwise indicated. Data are presented as means ± SEM, and with $P$-values < 0.05 were considered statistically significant. All mass spectrometry data were analyzed with MaxQuant (development version 1.2.7.1) All MS data were analyzed with MaxQuant (development version 1.2.7.1). R software, package corrplot [74] was used to analyze and plot proteomics data, gene expression survival analysis, and IHC correlation matrices. Further, statistical calculations and analysis involving standard deviations, SEM, and the two-tailed $t$-tests were conducted using excel spreadsheets. Image analysis was conducted using WCIF ImageJ software. An online calculator for Fisher exact test was also used, as described in the following website: http://www.socscistatistics.com/tests/fisher/Default2.aspx.

Kaplan–Meier curves were plotted with survival package [75], and *P*-values computed with a log-rank test, in R software. Gene expression analysis correlation and survival was computed on publicly available datasets, downloaded from Gene Expression Omnibus (GEO) of Ewing sarcoma (GSE63157) (Data ref: [76]) and osteosarcoma (GSE42352) (Data ref: [77]). Clinical data for the osteosarcoma cohort were obtained from (http://r2.amc.nl).

## Data availability

The mass spectrometry and proteomics data have been deposited to the ProteomeXchange Consortium via the PRIDE [78] partner repository, with the dataset identifier PXD014827. The data is available at (https://www.ebi.ac.uk/pride/archive/projects/PXD014827).

Expanded View for this article is available online.

## Acknowledgements

We thank Jennifer Baker, Andrew Minchinton, Vladimir Zeenko, Michael Lizardo, Amy Li, Sylvia Lee, Rizaldy Gamboa, and Rebecca Wu for reagents, technical assistance, and helpful discussions. AME is funded through a Michael Smith Foundation for Health Research (MSFHR) trainee award (Grant ID # 17159). This work was supported by funds from Terry Fox Research Institute Team Grant 1021 (to DH, GM, and PHS) and funds from CIHR Foundation grant FDN-143280 (to PHS). This research was also supported in part by a Terry Fox New Frontiers Program Project Grant #1062 to PHS.

## Author contributions

PHS and AME-N designed the study; AME-N conducted the majority of experiments; SPS performed initial studies with MS-275; YeW and DGH conducted cell-based screening; MP, XQW, BR, FZ, HZ, and JC provided extensive technical assistance; HC and YuW performed animal studies; GLN conducted bioinformatics analyses; AD, AME-N, and PHS performed pathology analysis; SC, AM, CSH, and GM conducted proteomic studies; NK and MG provided support and advised on critical experiments; DS and OD established and provided the EwS PDX; AME-N, GBM, and PHS wrote the manuscript with input from all authors.

## Conflict of interest

The authors declare that they have no conflict of interest.

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
