## [Review Process File · EMBO Reports]

Class I HDAC inhibitors enhance YB-1 acetylation and oxidative stress to block sarcoma metastasis

Amal M El-Naggar, Syam Prakash Somasekharan, Yemin Wang, Hongwei Cheng, Gian Luca Negri, Melvin Pan, Xue Qi Wang, Alberto Delaidelli, Bo Rafn, Jordan Cran, Fan Zhang, Haifeng Zhang, Shane Colborne, Martin Gleave, Anna Mandinova, Nancy Kedersha, Christopher S Hughes, Didier Surdez, Olivier Delattre, Yuzhuo Wang, David G Huntsman, Gregg B Morin, and Poul H Sorensen

Review timeline:

Submission date:	29 April 2019
Editorial Decision:	14 June 2019
Revision received:	2 August 2019
Editorial Decision:	20 September 2019
Revision received:	29 September 2019
Accepted:	6 October 2019

Editor: Achim Breiling

Transaction Report:

1st Editorial Decision

14 June 2019

Thank you for the transfer of your research manuscript to EMBO reports. We have now received reports from the three referees that were asked to evaluate your study, which can be found at the end of this email.

As you will see, all referees think that the findings are of interest, but they also have several comments, concerns and suggestions that need to be addressed to render the manuscript suitable for publication in EMBO reports. As the reports are below, I will not further detail them here, also as I feel that all points need to be addressed as indicated by the referees.

Given the constructive referee comments, we would like to invite you to revise your manuscript with the understanding that all referee concerns must be addressed in the revised manuscript and in a detailed point-by-point response. Acceptance of your manuscript will depend on a positive outcome of a second round of review. It is EMBO reports policy to allow a single round of revision only and acceptance or rejection of the manuscript will therefore depend on the completeness of your responses included in the next, final version of the manuscript.

Revised manuscripts should be submitted within three months of a request for revision; they will otherwise be treated as new submissions. Please contact me if a 3-months time frame is not sufficient so that we can discuss the revisions further.

I look forward to seeing a revised version of your manuscript when it is ready. Please let me know if you have questions or comments regarding the revision.

REFEREE REPORTS

Referee #1:

The study by El-Naggar et al. (Sorenson) addresses the controversy concerning whether anti-oxidation prevents or promotes cancer, and whether reactive oxygen species (ROS) promote or suppress tumorigenesis. The authors reasoned that since oxidative stress attenuates melanoma metastasis they explored this in Ewing and osteosarcoma using cell-based screens for several compounds to increase ROS in combination with high concentration of sodium arsenite to stimulate ROS and identified class I HDAC inhibitor MS-275. MS-275 was found to do more than just modify histones - blocks the expression of NRF2 through acetylation of YB-1, thereby preventing the ability of YB-1 to translationally activate NFE2L2, as well as G3BP1 and HIF1A. Overall, the findings are novel and reveal necessary insight into the childhood bone sarcomas. *in vivo* results show that MS-275 treatment decreases invasion and metastasis, increases oxidative stress, and the MS-275-resistant (K81A) mutant restored metastatic capacity, as well as NRF2, HIF1 α , and G3BP1 synthesis in MS-275 treated mice. The authors suggest the therapeutic possibility of targeting YB-1 with HDAC inhibitors in combination with other agents may be an effective strategy for sarcoma.

General comments:

While there is much about the manuscript that is important, as presented the text is often confusing, difficult to follow and needs restructuring. Abbreviations are often not described. Some supplemental figures are not correctly identified. In many cases it is not mentioned how many replicates were performed and immunoblots were not quantified. Color coding in some figures is confusing and the same color is sometimes used for different samples and sometimes not. At times lane numbers are mentioned in the text but are not shown in figures.

Major criticisms:

1. It is stated on page 7: "HDAC inhibitors are known to enhance ROS levels". In this study this effect never occurred and the studies for this manuscript were performed with very high concentrations of sodium arsenite to stimulate ROS. This discrepancy needs to be addressed. In fact, in Fig. 1C cells were incubated with 500 μ M of NaAsO₂, yet there was no increase in ROS levels. Moreover, since this a much higher dose than the sublethal dose of 100 μ M why was this even done?
2. Figure 1E: In addition to the issues raised above, how can the authors explain that at higher dose of NaAsO₂ (500 μ M) they don't see expression of cleaved PARP, when in Figure 1B there is an increase of active caspase 3/7 at lower dose of NaAsO₂ (100 μ M)?
3. Figure EV1A: How do the authors explain that MS-275 treatment alone increases ROS levels in CHLA10 cells but does not have any effect in U2OS cells (Figure 1C)? It would be expected that since MS-275 activates NADPH oxidase (NOX) and increases ROS levels, addition of NaAsO₂, should further increase ROS levels, as seen in the CHLA10 cells.
4. It is not clear whether the authors screened the ROS inducer compounds in more than one cell line.
5. Did the authors examine the effect of other two classes of HDAC inhibitors (Quisinostat, and Romidepsin) apart from MS-275 (Entinostat)? The authors justified the used of Entinostat in all their studies because it is on clinical trial. The other two compounds are also in phase 2 clinical trials. Verify: <https://clinicaltrials.gov/ct2/show/NCT01947140>
6. Figure 2: The authors should show the polysome profiles after treatment with MS-275 and or NaAsO₂.
7. Figure 2A and 2B: Is the decrease of NRF2 and ARE activity in cells treated with the combination of MS-275 and NaAsO₂ a result of upregulation of Keap1, which regulates NRF2 degradation?

8. Figure 2C: It is not clear how can the authors conclude that ectopic NRF2 expression blocks the ability of MS-275 to induce ROS accumulation, when the cells treated with MS-275 alone and MS-275 with NRF2 O/E are normalized. Unless this reviewer misunderstands the experiment, the only group that should be normalized is the vehicle untreated group. If ectopic NRF2 blocks the ability to induce ROS accumulation, then shouldn't it be comparing MS-275 alone vs. MS-275 with NRF2 O/E?

9. Figure EV2: The authors should explain why MS-275 treated cells have more relative NRF2 protein levels compared to vehicle, when in Figure 2A there is no change in NRF2 expression with MS-275 treatment.

10. Figure 3: The mass spectrometry data identified several MS-275 acetylation targets. The authors mentioned that they examined proteins involved in mRNA translation, however they did not comment on other key proteins including key translation initiation factors. Are there other important top targets?

11. Figure EV3B: The text refers to CHLA10 cells, but the figure caption states U2OS cells. This needs to be clarified. Did the authors expect to see G3BP1 acetylation under any condition? Where is the G3BP1 (input)?

12. Figure 4A: Under any condition, not only H₂O₂ or NaAsO₂, the silencing of YB-1 decreases expression of NRF2. According to results presented in Figure 4, MS-275 increases acetylation of YB-1 K81 to decrease NFE2L2 binding, in turn blocking NRF2 translation by YB-1. However, these results are seen only upon addition of NaAsO₂, meaning under oxidative stress. The authors do not show the results with MS-275 treatment alone which needs to be provided (Figure 4E).

13. Figure 5: MS-275 decreases HIF1alpha in CHLA10 cells under hypoxia (Fig 5A) and G3BP in U2OS cells under NaAsO₂ treatment (Fig 5B). In Figure 5C, MS-275 increases H1FA polysomes of CHLA10 cells under hypoxia (1% O₂) compared to ambient conditions (21% O₂). Additionally, MS-275 decreases G3BP1 polysomes in CHLA10 or U2SO cells, which are not under NaAsO₂ treatment. This is inconsistent with the data presented in Fig. 5B compared to Fig. 5D, and these are different cells, treatment conditions, and results. This all needs to be clarified.

14. Figure EV4C: We should be able to see expression of YB-1 in the empty vector (EV) sample treated with MS-275 and NaAsO₂ treatment, since this combination of treatments does not affect its expression (Fig. 5A and B). We also should be able to see expression of YB-1 in the WT YB-1, since it should be overexpressing YB-1, unless the combination of treatments is decreasing YB-1 expression (seen in all panels), and as a result is decreasing G3BP1 and SG formation. Again, this needs to be sorted out.

15. In the in vivo studies the authors do not mention what is the vehicle treatment.

16. Is the significance in Figure EV5C and Figure 6C due to the smaller number of mice in the MS-275 group compared to vehicle group?

17. Figure EV6G: The MS-275 treatment shows increased expression of SG markers (FMRP and TIA1) compared to vehicle in the images, but the addition of Draq, does not allow one to visualize the merged images. The authors should only merge the two SG markers images, to compare the cells displaying SGs.

18. Figure 8: The authors mentioned p.17: "we analyzed publicly available EwS and OS gene expression datasets, and found strong correlations between HIF1A, G3BP1, and NFE2L2 transcripts in EwS and OS HIF1A, G3BP1, and NFE2L2 transcripts in EwS and OS ". Given the shape of the scatter plot, the authors should consider richer or more complex models than those provided only by the Pearson correlation coefficients.

19. Animal studies: The lung lesions are very few (2 arrows). The number of animals is not the same between control and treatment, and it is not clear whether the compound MS-275 used was toxic.

20. Figure EV8: The authors do not show the expression of YB-1 in EwS or OS clinical samples. The authors should analyze the expression of YB-1 in samples and the co-expression of its targets.

Minor criticisms

- Some references are not updated- please revise new literature
- Indicate what DCFH-DA stands for
- Explain what is a GSH/GSSG ratio?
- Tevise the following sentences: "Mechanistically, MS-275 blocked expression of the master antioxidant factor, NRF2, which reduces cellular ROS, but not transcriptionally"
- Poul H Sorensen^{1,2,9}. There is not a # 9 note.

----- Referee #2:

This study provides an interesting series of experimental results, highlighting the relevance of a mechanism of control of oxidative stress response that is partially novel. Several points however require further understanding/better explanation-details:

- The rationale for the use of a sodium arsenite pulse in the cell screening is rather unclear. Why is this approach chosen? It is also difficult to assess the relevance/parallel among this part of the study, and later experiments (where sodium arsenite is not used). The authors show lack of cell death upon MS-275 treatment at 1 μ M (24h treatment). Is cell death observed at later time points?

- The authors start with a cell screening, where several other HDAC inhibitors had an effect similar to MS-275. They should evaluate whether the mechanism they proposed is HDAC inhibitor-specific.

- o Minor: the authors state that they chose to focus on MS-275 since it is in clinical trials for solid tumors, however other HDAC inhibitors are already approved and are being tested in solid tumors likewise....

- Sodium arsenite has been proposed to modulate histone acetylation (Ramirez T, Brocher J et al. Sodium arsenite modulates histone acetylation, histone deacetylase activity and HMGN protein dynamics in human cells. *Chromosoma*. 2008 Apr;117(2):147-57. Epub 2007 Nov 13. PubMed PMID: 17999076). Controls should be performed to check for this variable.

- The protein levels of NRF2 have been already described to change upon HDAC inhibition or down-regulation (either up-regulated or down regulated, depending on the cell type/experimental conditions: see as an example Mercado N, et al. Decreased histone deacetylase 2 impairs Nrf2 activation by oxidative stress. *Biochem Biophys Res Commun*. 2011 Mar 11;406(2):292-8. doi: 10.1016/j.bbrc.2011.02.035. Epub 2011 Feb 12. PubMed PMID: 21320471). It seems that the variability is quite high, and it is difficult to assess whether the results shown (limited to a small number of cell lines, and PDX models that are not described in sufficient details) have a limited, or rather a more general relevance. This is especially relevant since several other mechanisms have been proposed to mediate the ROS modulating effect of HDAC inhibition, and in many cases results have been shown, that are difficult to reconcile with the present study if not assuming that different mechanisms may take place, in different cells/different context (and then of course the study -though of interest- has a limited scope, without the dissection of the context required to trigger the mechanism identified).

- The experiments with wt vs mutant YB-1 are performed under over-expression conditions, at present times technologies are available to establish more reliable cell models by direct introduction of the mutation in the tumor cell genome.

----- Referee #3:

In this manuscript El-Naggar et al describe the mechanisms of the class I HDAC inhibitor MS-275 function in sarcoma. The authors specifically evaluate the impact on metastasis, a relevant problem in the clinical management of sarcoma patients where treatment and control of lung metastasis is

problematic. The manuscript is timely and highly relevant to the field. The study itself was performed thoroughly and is described well.

Major points

1. Page 7 line 4: Was the media changed between NaAsO₂ treatment and 24h incubation? The treatment conditions may affect compound stability.
2. Page 7 line 13: The data should be included into supplemental.
- 3.

Minor points

1. Figure 7A: Please add sizing bars to each image and increase resolution.
2. Methods: where were the cells purchased from?

1st Revision - authors' response

2 August 2019

REVIEWER COMMENTS:

Referee #1:

The study by El-Naggar et al. (Sorensen) addresses the controversy concerning whether anti-oxidation prevents or promotes cancer, and whether reactive oxygen species (ROS) promote or suppress tumorigenesis. The authors reasoned that since oxidative stress attenuates melanoma metastasis they explored this in Ewing and osteosarcoma using cell-base screens for several compounds to increase ROS in combination with high concentration of sodium arsenite to stimulate ROS and identified class I HDAC inhibitor MS-275. MS-275 was found to do more than just modify histones - blocks the expression of NRF2 through acetylation of YB-1, thereby preventing the ability of YB-1 to translationally activate NFE2L2, as well as G3BP1 and HIF1A. Overall, the findings are novel and reveal necessary insight into the childhood bone sarcomas. in vivo results show that MS-275 treatment decreases invasion and metastasis, increases oxidative stress, and the MS-275-resistant (K81A) mutant restored metastatic capacity, as well as NRF2, HIF1 α , and G3BP1 synthesis in MS-275 treated mice. The authors suggest the therapeutic possibility of targeting YB-1 with HDAC inhibitors in combination with other agents may be an effective strategy for sarcoma.

General comments:

While there is much about the manuscript that is important, as presented the text is often confusing, difficult to follow and needs restructuring. Abbreviations are often not described. Some supplemental figures are not correctly identified. In many cases it is not mentioned how many replicates were performed and immunoblots were not quantified. Color coding in some figures is confusing and the same color is sometimes used for different samples and sometimes not. At times lane numbers are mentioned in the text but are not shown in figures.

We thank the reviewer for the supportive comments. We hope that our revisions, in addition to addressing scientific concerns, have also made the manuscript more clear, such as correcting figure numbering, defining abbreviations, indicating the number of replicates (which was actually already detailed in the previous version in the figure legends), correcting color coding, and clarifying lane assignments. However, without more specific detail from the reviewer, it is difficult to know exactly which figures or text the reviewer is referring to.

Major criticisms:

1. *It is stated on page 7: "HDAC inhibitors are known to enhance ROS levels". In this study this effect never occurred and the studies for this manuscript were performed with very high concentrations of sodium arsenite to stimulate ROS. This discrepancy needs to be addressed. In fact, in Fig. 1C cells were incubated with 500 μ M of NaAsO₂, yet there was no increase in ROS levels. Moreover, since this a much higher dose than the sublethal dose of 100 μ M why was this even done?*

We thank the reviewer for giving us the opportunity explaining this critical point in more detail. Our screen was primarily focused on screening for compounds that block the anti-oxidant response in sarcoma cells, rather than only increasing ROS itself. We reasoned that if we added an exogenous

ROS inducer (in this case NaAsO₂), along with agents that block the cellular response to oxidative stress, this should increase cell death. We did not mean to imply that MS-275 or other HDAC inhibitors induce ROS on their own through unknown mechanisms to kill cells, but rather that they indirectly do so via their ability to block the oxidative stress response (as we show later in the manuscript is mediated by inhibiting YB-1 activity and thus NRF2 expression). We have now clarified these points on p.7, line 146-150, p.8, line 151, and have changed the title of the first section to “Cell-based screens identify MS-275 as an oxidative stress sensitizer in sarcoma cells” p.6, line 119, and on p.18, lines 399-400, and p.19, lines 401-404.

We chose 100 μ M NaAsO₂ as a sublethal dose (from our preliminary studies and the literature) for the screen, as we did not want to complicate the results by using lethal doses of NaAsO₂. The reviewer is correct in that **Fig. 1C** and a few other studies used a higher NaAsO₂ dose (500 μ M). The reason for using this higher dose is that a secondary endpoint of our screens was to identify agents that block stress granule formation. These cytosolic RNA/protein aggregates rapidly assemble under enhanced oxidative stress, including tumor cells (PMCID:PMC4993645 and PMCID: PMC4384734). We therefore reasoned that stress granules are a good secondary readout for our screen, particularly since they are relatively easy to detect in a fluorescent screen set-up. Since 500 μ M of NaAsO₂ is a well-established dose for inducing stress granules in the literature, we used 500 μ M of NaAsO₂ in **Fig. 1C**, and left that dose for **Fig. 1C** and few other figures, residually. Indeed, there is a small but significant increase in ROS accumulation when vehicle alone is compared to arsenite even in the absence of MS-275 ($p = 0.044$). However, we have now redone this experiment at 100 μ M NaAsO₂, as shown in new **Fig. EV1B**; we confirmed that 100 μ M NaAsO₂ also induces stress granules, as shown in **Fig. L1** of the response letter. Most of the other studies were later conducted under 100 μ M of NaAsO₂, such as previous **Figs. 2B-E, now Fig. 1G-J**. To avoid further confusion, we have now taken out the sentence “HDAC inhibitors are known to enhance ROS levels”.

2. Figure 1E: In addition to the issues raised above, how can the authors explain that at higher dose of NaAsO₂ (500 μ M) they don't see expression of cleaved PARP, when in Figure 1B there is an increase of active caspase 3/7 at lower dose of NaAsO₂ (100 μ M)?

We apologize for making an error in the **Fig. 1E** legend. The NaAsO₂ dose used for this experiment was actually 100 μ M, and this has been corrected in the corresponding figure legend. The apparent discrepancy between cleaved PARP by Western blotting and active caspase 3/7 assessment using an apoptosis kit is likely attributed to the difference in optimal timing or sensitivity between these two different techniques.

3. Figure EV1A: How do the authors explain that MS-275 treatment alone increases ROS levels in CHLA-10 cells but does not have any effect in U2OS cells (Figure 1C)? It would be expected that since MS-275 activates NADPH oxidase (NOX) and increases ROS levels, addition of NaAsO₂, should further increase ROS levels, as seen in the CHLA-10 cells.

We thank the reviewer for this point. There are many differences between U2OS and CHLA-10 cells, which might contribute to the reviewer's point. U2OS is an osteosarcoma cell line with functional p53 (wt), which is poorly tumorigenic and non-metastatic *in vivo* (PMCID: PMC4467467 and PMCID: PMC4552671). In contrast, CHLA-10 is a Ewing sarcoma cell line with non-functional p53 and is highly metastatic *in vivo* (PMID:25965573). Potentially CHLA-10 cells are under higher levels of basal oxidative stress, and therefore are less able to handle additional perturbations. These points plus other cell line differences likely explain the observed discrepancy the reviewer is alluding to. We were puzzled by the second comment: “MS-275 activates NADPH oxidase (NOX) and increases ROS levels”, as we did not refer to such results in our manuscript, nor could we find a reference to this relationship elsewhere in the literature. Therefore it is difficult to respond to this comment. However, we do observe that adding arsenite alone to U2OS cells alone significantly increases ROS compared to vehicle, albeit only moderately, and a corresponding p value has now been added to the figure legend of **Fig. 1C**. However, we observed a much more marked ROS increase in U2OS cells when arsenite was combined with MS-275 in **Fig. 1C**.

4. It is not clear whether the authors screened the ROS inducer compounds in more than one cell line.

The screen was conducted in U2OS cells as a standard sarcoma model with functional p53, and a well-established model for studying stress granule formation as well. However, MS-275 was also validated in a number of other cell lines including CHLA-10 (e.g **Fig. 2B** in the previous version,

now **Fig. 1G**) and other sarcoma cell lines, including MNNG, TC32, and A673, in experiments not included in the manuscript due to space limitations, and carcinomas (PC-3; letter **Fig. L2A-F**), where results were highly consistent with U2OS cells. As mentioned in the response to Reviewer 2, point 4, we prefer to keep the manuscript sarcoma-focused, also to keep within space limitations, and therefore have not added the PC3 data to the manuscript data package. However, this is an editorial decision and we can add this data if requested.

5. Did the authors examine the effect of other two classes of HDAC inhibitors (Quisinostat, and Romidepsin) apart from MS-275 (Entinostat)? The authors justified the used of Entinostat in all their studies because it is on clinical trial. The other two compounds are also in phase 2 clinical trials. Verify: <https://clinicaltrials.gov/ct2/show/NCT01947140>

The reviewer is correct in that other class I HDAC inhibitors found in our screen are also in clinical trials. Indeed, the use of Entinostat was chosen not only for being in trials but also because it showed the best toxicity profile in the screens, namely essentially no toxicity alone but high toxicity when combined with a 100 uM sublethal NaAsO₂ dose. We have changed the text to reflect this point on p.7, lines 139-150, and p.8, lines 151. However, as shown in new **Fig. EV2D**, we now report that Quisinostat and Romidepsin also enhance YB-1 acetylation at 1 uM. However, some cell toxicity was observed with these compounds alone between 6-12 hrs post-treatment (data not shown), and therefore we used MS-275 for further experiments. We have made this point more clearly in the manuscript, on p.7, lines 141-143. These compounds also block stress granule formation as shown in letter **Fig L3**.

6. Figure 2: The authors should show the polysome profiles after treatment with MS-275 and or NaAsO₂.

We have now performed this study as shown in **Fig. EV1H**, demonstrating polysome profiles in U2OS cells under different treatment conditions, as requested by the reviewer. These experiments show that while MS-275 fails to reduce polysome formation (i.e. does not block translation), NaAsO₂ does so (i.e. reduced polysomes and increased 80S formation), regardless of MS-275 addition. We have now also added this point on p.9, lanes 187-190.

7. Figure 2A and 2B: Is the decrease of NRF2 and ARE activity in cells treated with the combination of MS-275 and NaAsO₂ a result of upregulation of Keap1, which regulates NRF2 degradation?

We previously could not detect any appreciable Keap1 expression in these cell lines by Western blotting, with or without MS-275 treatment (data not shown). Moreover, we could indirectly rule out this possibility by standard cycloheximide pulse-chase protein degradation studies, which is already included in the manuscript as new **Fig. EV11-J (Fig. EV2A and B in previous version)**. This clearly showed no difference in NRF2 stability +/- MS-275, thus strongly arguing against a role for Keap1 on NRF2 stability as an explanation for the MS-275 effects. Furthermore, we used the MG132 proteosomal inhibitor as shown in letter **Fig L4**, which did not rescue NRF2 expression in the presence of MS-275 treatment, further arguing against reduced protein stability to explain MS-275 effects on NRF2.

8. Figure 2C: It is not clear how can the authors conclude that ectopic NRF2 expression blocks the ability of MS-275 to induce ROS accumulation, when the cells treated with MS-275 alone and MS-275 with NRF2 O/E are normalized. Unless this reviewer misunderstands the experiment, the only group that should be normalized is the vehicle untreated group. If ectopic NRF2 blocks the ability to induce ROS accumulation, then shouldn't it be comparing MS-275 alone vs. MS-275 with NRF2 O/E?

We thank the reviewer for pointing out this concern. As mentioned in our responses to points 1 and 3 of Reviewer 1, our model is that while MS-275 alone fails to appreciably increase ROS, when combined with a ROS inducing agent such as NaAsO₂, MS-275-treated cells show significantly higher ROS levels and increased cell death than cells treated with NaAsO₂ alone (i.e. that while MS-275 increases sensitivity to NaAsO₂, NRF2 overexpression blocks this ROS increase). In other words, in previous **Fig. 2C**, we wanted to show how ectopic NRF2 expression in cells reduces ROS levels in cells treated with *both* MS-275 and NaAsO₂ together. Therefore while columns 1, 2 show ROS increases under NaAsO₂ treatment alone, and columns 3 and 4 show significantly higher induction of ROS by adding MS-275 in the presence of NaAsO₂, shown in previous figures, the key new data in columns 5 and 6 clearly shows that NRF2 overexpression blocks ROS induction under MS-275 treatment in cells treated with NaAsO₂. Therefore, we believe the normalization procedure

we have used is correct. However, we have now simplified the figure as new **Fig. 1H** in the manuscript to address the reviewer's concern.

9. *Figure EV2: The authors should explain why MS-275 treated cells have more relative NRF2 protein levels compared to vehicle, when in Figure 2A there is no change in NRF2 expression with MS-275 treatment.*

In previous **Fig. EV2** (now **Fig EV1I and J**), cells were pre-treated with L-sulforaphane (SFN), a well-established agent for inducing NRF2, for several hours prior to MS-275 and cycloheximide (CHX) treatments, since NRF2 was only weakly or not expressed under ambient, non-oxidative stress conditions. SFN treatment was continued during CHX treatment. Therefore, the 0 time point actually reflects 4 hr post-SFN treatment. This is why there appears to be similar if not more NRF2 expression in MS-275 versus vehicle treated cells. We have now clarified this experimental detail in the figure legend. In contrast, in previous **Fig. 2A** (now **Fig. 1F**), Western blots show only the endogenous expression of NRF2; i.e. there is no SFN pre-treatment. NRF2 is induced under arsenite stress (2nd lane from the left) and this was virtually completely blocked by MS-275 (4th lane from the left).

10. *Figure 3: The mass spectrometry data identified several MS-275 acetylation targets. The authors mentioned that they examined proteins involved in mRNA translation, however they did not comment on other key proteins including key translation initiation factors. Are there other important top targets?*

We agree that there is a lot of potentially interesting data in previous **Fig 3** (now **Fig 2**). However, apart from histones and several ribosomal proteins, only DEK (DEK proto-oncogene), NAT10 (N-acetyltransferase 10), PARP1 (polyADP-ribose polymerase 1), TOP1 (DNA topoisomerase I), and YBX1 showed consistently increased acetylation under MS-275 treatment. We have added a sentence to mention this on p. 10, lines 208-210. We do not exclude the possible importance of other factors, especially with extended periods of treatment, but we focused on YB-1, as to follow up on other targets is beyond the scope of this manuscript.

11. *Figure EV3B: The text refers to CHLA-10 cells, but the figure caption states U2OS cells. This needs to be clarified. Did the authors expect to see G3BP1 acetylation under any condition? Where is the G3BP1 (input)?*

We apologize for this error, as the experiment was using U2OS cells, and we have corrected this in the manuscript. We tested G3BP1 several times and no acetylation was detected. We have now added the G3BP1 input to the figure, now **Fig EV2B**.

12. *Figure 4A: Under any condition, not only H2O2 or NaAsO2, the silencing of YB-1 decreases expression of NRF2. According to results presented in Figure 4, MS-275 increases acetylation of YB-1 K81 to decrease NFE2L2 binding, in turn blocking NRF2 translation by YB-1. However, these results are seen only upon addition of NaAsO2, meaning under oxidative stress. The authors do not show the results with MS-275 treatment alone which needs to be provided (Figure 4E).*

We agree that YB-1 inactivation likely reduces NRF2 expression translationally under diverse conditions, as YB-1 drives *NFE2L2* translation, based on our studies. The reviewer is requesting us to test MS-275 effects on NRF2 expression under non-oxidative stress conditions, which is a bit artificial, given that NRF2 is mainly induced under oxidative stress (although not exclusively). Regardless, we have now performed that experiment, new letter **Fig L5**.

13. *Figure 5: MS-275 decreases HIF1alpha in CHLA-10 cells under hypoxia (Fig 5A) and G3BP in U2OS cells under NaAsO2 treatment (Fig 5B). In Figure 5C, MS-275 increases H1FA polysomes of CHLA-10 cells under hypoxia (1% O2) compared to ambient conditions (21% O2). Additionally, MS-275 decreases G3BP1 polysomes in CHLA10 or U2SO cells, which are not under NaAsO2 treatment. This is inconsistent with the data presented in Fig. 5B compared to Fig. 5D, and these are different cells, treatment conditions, and results. This all needs to be clarified.*

We apologize for several labelling errors that may have contributed to the reviewer's concerns. All of the studies in these figures were performed in CHLA-10 cells. For the *G3BP1* studies, we inadvertently neglected to mention in previous **Fig 5D** that the experiment was conducted under 100 uM of arsenite (i.e. oxidative stress). The figure legend (now **Fig 4D**) is now fixed to reflect this point. Therefore MS-275 reduces *G3BP1* polysome association (and therefore *G3BP1* protein synthesis) under oxidative stress, as we claim in the text. For *H1FA*, we inadvertently mixed up the

polysome data, which has now been fixed in new **Fig 4C**, showing clearly that MS-275 also decreases polysome association of *HIF1A* under hypoxia. It is not surprising that these two YB-1 stress-related translational targets, G3BP1 and HIF1a, are each translationally regulated under their respective context-specific stress forms. For example, it would be out of the scope of our work to be expected to measure G3BP1 translation under the same stress (i.e. hypoxia) as for HIF1a.

14. *Figure EV4C: We should be able to see expression of YB-1 in the empty vector (EV) sample treated with MS-275 and NaAsO₂ treatment, since this combination of treatments does not affect its expression (Fig. 5A and B). We also should be able to see expression of YB-1 in the WT YB-1, since it should be overexpressing YB-1, unless the combination of treatments is decreasing YB-1 expression (seen in all panels), and as a result is decreasing G3BP1 and SG formation. Again, this needs to be sorted out.*

The reviewer is correct, and we have therefore selected more representative images in a new **Fig EV3C**, clearly showing YB-1 expression in the aforementioned panels.

15. *In the in vivo studies the authors do not mention what is the vehicle treatment.*

We apologize for this omission. The vehicle solution consists of 0.05 N HCl and 0.1% Tween 80 in saline solution, which is now added to the Materials and Methods section; p.23, lines 506-507

16. *Is the significance in Figure EV5C and Figure 6C due to the smaller number of mice in the MS-275 group compared to vehicle group?*

We strongly believe that the observed effects are not attributable to the difference in sample size. In previous **Fig 6C-D**, now **Fig 5C-D**, one mouse died due to non-tumor causes just prior to the experimental end point. Extensive histopathological examination failed to show any evidence of lung metastases in this MS-275 treated mouse. However, we choose not to include this mouse, given its death prior to experimental end point, even though it would have made our results even more convincing. As shown, the differences are already statistically significant.

The same finding is observed in previous **Fig EV5C**, now **Fig EV4C**, where despite the number of mice (again due to non-tumor related deaths prior to endpoint), there are significantly fewer mice with lung metastases in the MS-275 treated cohort. Moreover, in previous **Fig 7A**, now **Fig 6B**, the EV and wt-YB-1 mouse cohorts both showed dramatically reduced lung metastases under MS-275 treatment, compared to the vehicle treated cohort. Taken together, this strongly argues against the possibility that the observed phenotype and effects on metastasis are attributed to varying mice numbers. Furthermore, tumor lysates confirmed significant downregulation of critical YB-1 regulated factors promoting metastasis in previous **Fig 7C**, now **Fig 6E**, and previous **Fig EV6E-F**, now **Fig EV5E-F**.

17. *Figure EV6G: The MS-275 treatment shows increased expression of SG markers (FMRP and TIA1) compared to vehicle in the images, but the addition of Draq, does not allow one to visualize the merged images. The authors should only merge the two SG markers images, to compare the cells displaying SGs.*

We added new representative images for this figure, taking out the DRAQ images, and merged only the two SG marker panels as requested. This is now included in new **Fig EV5G**. MS-275 treatment, again showing significantly reduced SG formation. We failed to observe any increase in either FMRP or TIA1 levels by Western blotting in lysates from the same samples, as shown in letter **Fig L6**.

18. *Figure 8: The authors mentioned p.17: "we analyzed publicly available EwS and OS gene expression datasets, and found strong correlations between HIF1A, G3BP1, and NFE2L2 transcripts in EwS and OS HIF1A, G3BP1, and NFE2L2 transcripts in EwS and OS ". Given the shape of the scatter plot, the authors should consider richer or more complex models than those provided only by the Pearson correlation coefficients.*

We agree with the reviewer that the observed correlations between gene expression profiles are not completely linear. We now report Spearman's ρ instead of Pearson correlations in new **Fig 7** of the manuscript to account for non-linearity. While non-monotonic models could potentially fit the data better, their significance is harder to reconcile with actual biological meaning. The relationship between the expression profiles of these three genes is likely the result of complex gene regulations networks, the unraveling of which is beyond the scope of this study.

19. *Animal studies: The lung lesions are very few (2 arrows). The number of animals is not the same between control and treatment, and it is not clear whether the compound MS-275 used was toxic.* This seems to be similar to point 16. The variation in mouse numbers is attributed, as previously stated in the manuscript, to non-tumor deaths, such as failure to recover from surgery. This is not something that we can influence, other than spending thousands of dollars on repeated experiments to get the numbers to be exactly the same (which would be against animal ethics protocols). The results of several individual experiments are consistent and the sum of many included experiments provide compelling evidence for our conclusions. In regards to MS-275, the compound is non-toxic as shown and reported in multiple published *in vivo* studies (e.g. PMID: PMC4136565; PMID: PMC6398866; PMID: PMC5366870), and others. For lung metastases, we are presenting the most representative sections, and cannot be expected to include every single lung section that we analyzed. We have now added additional H&E images in letter **Fig L7** for the reviewer.

20. *Figure EV8: The authors do not show the expression of YB-1 in EwS or OS clinical samples. The authors should analyze the expression of YB-1 in samples and the co-expression of its targets.* We thank the reviewer for this suggestion. We now provide YB-1 expression as well in **Fig 8B** (Ewing sarcoma) and **Fig 8C** (osteosarcoma). YB-1 is strongly expressed across both sarcoma types, as we have previously reported (PMID:25965573). Moreover, in novel data in new **Fig 8B-C**, we show very strong correlations between YB-1 and NRF2, G3BP1, and HIF1a at the protein level. This clearly demonstrates the link between these proteins in sarcoma tissues.

Minor criticisms

- *Some references are not updated- please revise new literature*

We have now provided updated references; p.3, line 40 and p.5, line 99.

- *Indicate what DCFH-DA stands for*

We have included a definition of (CM- H₂DCFDA), commonly known as DCFDA, or chloromethyl derivative of 2',7' -dichlorofluorescein diacetate; p.7, line 147.

- *Explain what is a GSH/GSSG ratio?*

Glutathione (g-glutamylcysteinylglycine), a major cellular anti-oxidant, exists in reduced (GSH) and oxidized (GSSG) states. Therefore, the GSH/GSSG ratio is a useful and well-established indicator of oxidative stress in cells and tissues (i.e. the lower the ratio, the more oxidative stress is present as GSH becomes oxidized to GSSG). We have now included in p.8, lines 153-154, and p.31, lines 674-678.

- *Revise the following sentences: "Mechanistically, MS-275 blocked expression of the master antioxidant factor, NRF2, which reduces cellular ROS, but not transcriptionally"*

This has now been revised on p. 2, lines 26-28, as "Mechanistically, MS-275 inhibits YB-1 deacetylation, decreasing physical binding between YB-1 and the 5UTR of *NFE2L2*, thereby non-transcriptionally reducing translation and expression of the master antioxidant factor, NRF2, which reduces cellular ROS."

- *Poul H Sorensen1,2,9. There is not a # 9 note.*

The #9 note has been removed.

Referee #2:

This study provides an interesting series of experimental results, highlighting the relevance of a mechanism of control of oxidative stress response that is partially novel. Several points however require further understanding/better explanation-details:

We thank the reviewer for the supportive comments.

1. *The rationale for the use of a sodium arsenite pulse in the cell screening is rather unclear. Why is this approach chosen? It is also difficult to assess the relevance/parallel among this part of the study, and later experiments (where sodium arsenite is not used). The authors show lack of cell death upon MS-275 treatment at 1 μM (24h treatment). Is cell death observed at later time points?*

This comment is similar to Referee 1, point 1. Please see the response to that comment. To reiterate, the screen was based on the hypothesis that since oxidative stress blocks melanoma metastasis, blocking adaptation to oxidative stress might similarly reduce metastatic capacity in EwS and OS. We reasoned that combining a pulse of NaAsO₂ as an exogenous ROS inducer with agents that

block the anti-oxidant response should increase cell death, which is what the screen was set up to measure. This allowed us to identify MS-275 (and other class I HDAC inhibitors) as blocking the oxidative stress response by inhibiting YB-1 activity and thus NRF2 expression. We have now clarified these points on p.7, line 146-150, p.8, line 151, and to make this more understandable we changed the title of the first section to “Cell-based screens identify MS-275 as an oxidative stress sensitizer in sarcoma cells” p.6, line 119, and on p.18, lines 399-400, and p.19, lines 401-404.

The use of NaAsO₂ was based on its well-established ability to acutely induce ROS. More chronic ROS induction would likely lead to many secondary effects, which is why we chose 1-2 hr pulses. Regarding longer time points of MS-275 treatment alone, we tested 48-72 hrs with U2OS cells, but did not observe any appreciable cell death. Only at much high MS-275 concentrations (e.g. 5 or 10 uM) was significant cell death observed (see letter **Fig L8**). In later studies, NaAsO₂ was not used, as it is not generally applicable to *in vivo* studies, and is not appropriate for *in vitro* hypoxia studies.

2. The authors start with a cell screening, where several other HDAC inhibitors had an effect similar to MS-275. They should evaluate whether the mechanism they proposed is HDAC inhibitor-specific.

This is an important point, and is similar to Reviewer 1, point 5. We therefore tested two other class I HDAC inhibitors including Quisinostat and Romidepsin. As shown in new **Fig EV2D** of the manuscript, YB-1 acetylation is also markedly enhanced by treatment with the aforementioned agents, similar to MS-275 treatment. Likewise, these compounds both inhibit SG formation as shown in letter **Fig L3**. Please also refer to our response to Referee1, point 5.

o Minor: the authors state that they chose to focus on MS-275 since it is in clinical trials for solid tumors, however other HDAC inhibitors are already approved and are being tested in solid tumors likewise....

The reviewer is correct in that this is not, by itself, adequate justification for choosing MS-275 for further studies. As mentioned in the response to Reviewer 1, point 5, MS-275 was also chosen because it showed no or minimal toxicity alone, even under longer periods of treatment; see **letter Fig L8**, while inducing significant cell death in combination with NaAsO₂. As mentioned in the above paragraph, we now report that Quisinostat and Romidepsin also enhance YB-1 acetylation at 1 uM. However, some cell toxicity was observed with these compounds alone between 6-12 hrs post-treatment (data not shown), and therefore we used MS-275 for further experiments. We have made this point more clearly in the manuscript, on p.7, lines 139-150 and p.8, lines 151.

3. Sodium arsenite has been proposed to modulate histone acetylation (Ramirez T, Brocher J et al. Sodium arsenite modulates histone acetylation, histone deacetylase activity and HMGN protein dynamics in human cells. Chromosoma. 2008 Apr;117(2):147-57. Epub 2007 Nov 13. PubMed PMID: 17999076). Controls should be performed to check for this variable.

We thank the reviewer for this valid concern. Indeed, in the aforementioned study, NaAsO₂ was used for 24 hrs, much longer than our 1 hr pulses, and therefore potentially sufficient to induce indirect epigenetic modifications. Moreover, as far as we know, this study has not been reproduced in the literature. However, to explore this point, we performed additional immunoprecipitation using bead-conjugated anti-acetyl-lysine antibodies and U2OS cells exposed to different conditions (vehicle, NaAsO₂ at 500 uM, MS-275, and MS-275 + NaAsO₂), followed by Western blotting using unconjugated pan-acetyllysine antibodies. No significant changes in overall protein acetylation was observed with NaAsO₂ alone, even at 500 uM concentrations of the latter, compared with vehicle treatment (see letter **Fig L9A-B**); on the other hand, MS-275 increased overall acetylation as expected, but combined MS-275/ NaAsO₂ treatment showed relatively less acetylation compared to MS-275 alone. We realize that we did not specifically test histone acetylation in these studies, but we conclude that we do not see evidence that NaAsO₂ induces appreciable protein acetylation on its own in sarcoma cells.

4. The protein levels of NRF2 have been already described to change upon HDAC inhibition or down-regulation (either up-regulated or down regulated, depending on the cell type/experimental conditions: see as an example Mercado N, et al. Decreased histone deacetylase 2 impairs Nrf2 activation by oxidative stress. Biochem Biophys Res Commun. 2011 Mar 11;406(2):292-8. doi: 10.1016/j.bbrc.2011.02.035. Epub 2011 Feb 12. PubMed PMID: 21320471). It seems that the variability is quite high, and it is difficult to assess whether the results shown (limited to a small number of cell lines, and PDX models that are not described in sufficient details) have a limited, or

rather a more general relevance. This is especially relevant since several other mechanisms have been proposed to mediate the ROS modulating effect of HDAC inhibition, and in many cases results have been shown, that are difficult to reconcile with the present study if not assuming that different mechanisms may take place, in different cells/different context (and then of course the study -though of interest- has a limited scope, without the dissection of the context required to trigger the mechanism identified).

We would like to emphasize that the important and novel aspects of this manuscript are that we characterize *NFE2L2* mRNA as a direct YB-1 target for translational activation under oxidative stress in sarcoma cells, which has never been reported, and we go on to show that this can be blocked by HDAC inhibitor-mediated YB-1 acetylation *in vitro* and *in vivo*. We show this in both EwS and OS cell lines. The article that the reviewer mentions is focused on bronchial epithelial cells in the context of COPD, and claims that blocking HDAC2 activity reduces the stability of NRF2. We clearly show that MS-275 does NOT affect NRF2 stability using cycloheximide pulse chase experiments (we refer the reviewer to **Fig EV2A-B** of the previous manuscript, now **Fig EV11-J**), and furthermore we now show that the proteasome inhibitor MG132 does not rescue NRF2 expression in the presence of MS-275 (see letter **Fig L4**). Therefore, we respectfully argue that MS-275 and likely other class I HDAC inhibitors do not affect NRF2 stability in sarcoma cells, and we instead provide a completely new mechanism for this process.

To further address the reviewer's comment regarding PDX models, we now provide much more detail on the Ewing sarcoma PDX model used in this study, on p. 23, lines 511-514 and p. 24, line 515 of the Materials and Methods section.

While elucidating whether this mechanism is also functioning in non-sarcoma cells is clearly beyond the scope of the current study, we have now conducted similar *in vitro* and *in vivo* studies using PC-3 prostate carcinoma cells. This demonstrated that MS-275 decreases PC-3 renal subcapsular xenograft *in vivo* local invasion (letter **Fig L2Ci-ii and L2D**) and lung metastasis (letter **Fig L2E**), and *in vitro* stress granule formation (letter **Fig L2F**). This strongly suggests a generalized effect rather than cell specific/tumor specific effects. Given that this manuscript is sarcoma focused, we have not put this data into the manuscript to keep it concentrated on sarcomas, and for the sake of brevity, but consider the potential inclusion of this data an editorial decision, and we are happy to put it in if requested.

5. The experiments with wt vs mutant YB-1 are performed under over-expression conditions; at present times technologies are available to establish more reliable cell models by direct introduction of the mutation in the tumor cell genome.

We agree that it would be more elegant to use a CRISPR approach to directly knock in the YB-1-K81A mutation into the endogenous locus of sarcoma cells. However, we tried unsuccessfully many times to knock out YB-1 in Ewing sarcoma cells using CRISPR technology. This may be explained by YB-1 being an essential gene in Ewing sarcoma cells (and potentially in osteosarcoma cells), as shown in the DepMap project of the Broad Institute (<https://depmap.org/portal/gene/YBX1>) using CRISPR screens (see also letter **Fig L10**). Indeed, our attempts to introduce mutations this way have unfortunately been unsuccessful, making the use of an overexpression system a necessity. We therefore did not feel that it is reasonable to expect us to spend up to a year perfecting this approach for the revised manuscript, as we already show significant results and a CRISPR based approach would not change our overall conclusions. Having said this, there are some advantages of the overexpression system. For example, MS-275 was still able to block *in vivo* local invasion and lung metastasis of sarcoma cells even with *wt* YB-1 overexpression.

Referee #3:

In this manuscript El-Naggar et al describe the mechanisms of the class I HDAC inhibitor MS-275 function in sarcoma. The authors specifically evaluate the impact on metastasis, a relevant problem in the clinical management of sarcoma patients where treatment and control of lung metastasis is problematic. The manuscript is timely and highly relevant to the field. The study itself was performed thoroughly and is described well.

We thank the reviewer for their encouraging and supportive comments.

Major points

1. *Page 7 line 4: Was the media changed between NaAsO₂ treatment and 24h incubation? The treatment conditions may affect compound stability.*

We wish to explain this point better. U2OS OS cells were pre-treated for 24 hrs with the indicated library at 1 μ M. The last one hr of the 24 hr drug treatment, a sublethal dose of 100 μ M NaAsO₂ was conducted. Then cells were washed (thus removing NaAsO₂) and fresh medium was added along with fresh compound treatments and NucView-488 dye, which binds to activated caspase-3/7 for monitoring cell apoptosis by Incucyte live cell monitoring for an additional 12 hrs. We have now clarified the screen procedure in more detail on p. 25, lines 539-550. Small scale *in vitro* experiments were also conducted, which did not show appreciable differences if we changed the medium before arsenite treatment or not, reflecting good stability of the compound. Further, acetylomic data showed histone acetylation, as a readout of MS-275 activity, in response to MS-275 treatment alone or MS-275 plus arsenite following the same treatment course used for the screen.

2. *Page 7 line 13: The data should be included into supplemental.*

We have now added this data to the supplemental figures as **Fig EV1A**.

Minor points

1. *Figure 7A: Please add sizing bars to each image and increase resolution.*

Scale bars representing 100 μ m are now added to each panel.

2. *Methods: where were the cells purchased from?*

The CHLA-10 Ewing sarcoma cell line was kindly provided by Dr. Patrick Reynolds (Texas Tech University). The U2OS cell lines was obtained from the American Type Culture Collection (ATCC). We have now added this information to the Materials and Methods section; p.24, lines 532-534.

References:

- 1 Ho, J., Tumkaya, T., Aryal, S., Choi, H. & Claridge-Chang, A. Moving beyond P values: data analysis with estimation graphics. *Nat Methods* **16**, 565-566, doi:10.1038/s41592-019-0470-3 (2019).
- 2 Tsherniak, A. *et al.* Defining a Cancer Dependency Map. *Cell* **170**, 564-576 e516, doi:10.1016/j.cell.2017.06.010 (2017).

2nd Editorial Decision

20 September 2019

Thank you for the submission of your revised manuscript to our editorial offices. We have now received the reports from the two referees that were asked to re-evaluate your study, you will find below. As you will see, both referees support the publication of your study in EMBO reports.

Before we can proceed with formal acceptance, I have these editorial requests that need to be addressed in a final revision:

- Please provide the abstract written in present tense (with not more than 175 words).
- Please add up to 5 keywords to the title page of the manuscript.
- Please format the references according to our journal style. See: <http://www.embopress.org/page/journal/14693178/authorguide#referencesformat>
- Please enter your funding information into our submission system upon final submission.
- Per journal policy, we do not allow 'data not shown', which is stated several times in the manuscript. All data referred to in the paper should be displayed in the main or Expanded View figures, or the Appendix. Thus, please add these data (or change the text accordingly, if these data are not important). See: <http://embor.embopress.org/authorguide#unpublisheddata>

- In the author contributions there are two authors mentioned as YW. Please differentiate (maybe using YeW and YuW).
- Thanks a lot for providing the detailed WB source data. However, as the source data will be linked to the figures in the online version of the paper, we need single separate files with the source data for each figure (main and EV). Thus, please upload all the source data for one figure as single pdf file.
- Some WB panels are over-contrasted, of low resolution, or panels within one figure show very different contrast and brightness. Please show the WBs with panels with equal contrast, and as unmodified as possible, and of best resolution possible.
- We require that all corresponding authors supply an ORCID ID for their name. We will not proceed before this is done. Please find instructions on how to link your ORCID ID to your account in our manuscript tracking system in our Author guidelines:
<http://www.embopress.org/page/journal/14693178/authorguide#authorshipguidelines>
- Finally, please find attached a word file of the manuscript text (provided by our publisher) with changes we ask you to include in your final manuscript text, and some queries, we ask you to address. Please provide your final manuscript file with track changes, in order that we can see the modifications done.

REFEREE REPORTS

Referee #1:

I have now carefully reviewed the revised manuscript. The authors have adequately addressed all of the concerns I raised in the previous review.

Referee #2:

The authors have answered to my concerns. Though several issues remain open. in my opinion there are several original data of interest for the field.

2nd Revision - authors' response

29 September 2019

The authors performed all minor editorial changes.

Corresponding Author Name: Poul H Sorensen

Manuscript Number: EMBOR-2019-48375